# Factored Value Functions for Graph-Based Multi-Agent Reinforcement Learning

Ahmed Rashwan[1]   Keith Briggs[2]   Chris Budd[1]   Lisa Kreusser[3]

## Abstract

Credit assignment is a core challenge in multi-agent reinforcement learning (MARL), especially in large-scale systems with structured, local interactions. Graph-based Markov decision processes (GMDPs) capture such settings via an influence graph, but standard critics are poorly aligned with this structure: global value functions provide weak per-agent learning signals, while existing local constructions can be difficult to estimate and ill-behaved in infinite-horizon settings. We introduce the *Diffusion Value Function (DVF)*, a factored value function for GMDPs that assigns to each agent a value component by diffusing rewards over the influence graph with temporal discounting and spatial attenuation. We show that DVF is well-defined, admits a Bellman fixed point, and decomposes the global discounted value via an averaging property. DVF can be used as a drop-in critic in standard RL algorithms and estimated scalably with graph neural networks. Building on DVF, we propose *Diffusion A2C (DA2C)* and a sparse message-passing actor, *Learned DropEdge GNN (LD-GNN)*, for learning decentralised algorithms under communication costs. Across the firefighting benchmark and three distributed computation tasks (vector graph colouring and two transmit power optimisation problems), DA2C consistently outperforms local and global critic baselines, improving average reward by up to $11\%$.

## 1. Introduction

Many large-scale multi-agent systems exhibit local interactions that can be represented by a graph, including communication networks (Lhazmir et al., 2020), power and smart grids (Angelidakis and Chalkiadakis, 2015), and recommender systems (Tavakol and Brefeld, 2014). In these settings, each agent influences only a limited set of neighbours, motivating *Graph-based Markov Decision Processes (GMDPs)* (Amato and Oliehoek, 2015; Nair et al., 2005), in which state transitions and rewards factor according to an influence graph. This structure enables scalable learning and, importantly, generalisation across graph sizes and topologies, a challenge that has recently been highlighted in cooperative graph-structured MARL benchmarks (Sintes and Busic, 2026; Liu et al., 2024).

Most MARL algorithms rely on a value function (critic) to estimate returns and provide a learning signal for policy optimisation. While GMDPs naturally admit factored value functions that exploit locality (Schneider et al., 1999), designing critics that are both informative and scalable remains difficult. Global value functions aggregate rewards across all agents and can become nearly insensitive to any single agent as the system grows, leading to weak credit assignment (Foerster et al., 2018). Local alternatives based on discounted sums of influenced rewards are more aligned with the influence graph, but can be difficult to estimate and may behave poorly in infinite-horizon settings (Oliehoek et al., 2008). Meanwhile, popular value decomposition methods such as VDN (Sunehag et al., 2018), QMIX (Rashid et al., 2020), DCG (Boehmer et al., 2020), and DFQ (Wang and Ke, 2022) factorise critic *architectures* to approximate a global value from a global reward, rather than defining a value function that explicitly reflects the GMDP influence structure. These limitations motivate a critic that is structurally aligned with GMDPs and practical for large-scale reinforcement learning.

To address this gap, we introduce the *Diffusion Value Function (DVF)*, a factored value function tailored to GMDPs. The DVF models how the effect of an agent's actions diffuses through the influence graph, placing greater weight on rewards that are temporally and spatially close. This yields a value function that is well-defined for infinite-horizon problems, can be learned with temporal-difference methods, and satisfies a decomposition property: the component-wise mean of the DVF recovers the global discounted value. We estimate diffusion values using a graph

---

[1]University of Bath [2]BT Research [3]Monumo. Correspondence to: Ahmed Rashwan <ar3009@bath.ac.uk>.

neural network (GNN) (Battaglia et al., 2018; Scarselli et al., 2008), yielding scalable value estimation and leveraging the locality bias of GNNs (Topping et al., 2022). DVF applies to any cooperative MARL task with local interactions that admit a GMDP-style factorisation, and can be used wherever a value function is required. In this work we adopt centralised training with decentralised execution (CTDE), but the locality of DVF also suggests a path toward decentralised training with neighbour communication, which we leave to future work.

Since value estimation underpins most MARL algorithms, DVF can be used as a drop-in critic for GMDP-based reinforcement learning. We instantiate this idea by integrating DVF into Advantage Actor–Critic (A2C) (Mnih et al., 2016), yielding a method we call *Diffusion A2C (DA2C)*.

We evaluate DA2C on cooperative graph-structured benchmarks, including the firefighting problem from (Oliehoek et al., 2008; Sintes and Busic, 2026), and on learning decentralised message-passing algorithms, where agents exchange information with neighbours to achieve a common objective (Barenboim and Elkin, 2022; Vatter et al., 2023; Foerster et al., 2016). In many distributed systems, communication is costly (e.g., due to interference or collisions), motivating policies that learn when to communicate in addition to what to compute (Baccelli et al., 2006; Linial, 1987). We therefore introduce a GNN-based actor, the *Learned DropEdge GNN (LD-GNN)*, which jointly learns sparse message passing and local outputs. Across vector graph colouring and two transmit power optimisation tasks, DA2C consistently outperforms both local and global critic baselines, improving average reward by up to 11%.

**Contributions.**

1. We introduce the *Diffusion Value Function (DVF)*, a factored value function for Graph-based MDPs that models the decay of influence over time and graph distance. We establish that DVF is well-defined in infinite-horizon settings (existence and uniqueness as a Bellman fixed point), decomposes the global value via an averaging property, and is aligned with the global objective in the sense that improving all diffusion components improves the global value.

2. We propose *Diffusion A2C (DA2C)*, an actor–critic algorithm that replaces the standard critic with DVF, yielding a scalable learning signal for cooperative MARL with local interactions and enabling generalisation across graph sizes and topologies.

3. We introduce the *Learned DropEdge GNN (LD-GNN)*, an actor architecture that learns sparse message passing under distributed-system constraints. Across three distributed computation benchmarks, DA2C

with LD-GNN consistently outperforms local and global critic baselines, improving average reward by up to 11%.

## 2. Graph-based Markov decision process

We consider multi-agent Markov decision processes (MDPs) whose structure is described by an *influence graph* (IG), following (Bagnell and Ng, 2005; Schneider et al., 1999). A multi-agent MDP is given by

$$\mathcal{M} = (\mathcal{V}, \mathcal{S}, \mathcal{A}, T, \mathcal{R}),$$

where $\mathcal{V} = \{1, \ldots, n\}$ indexes the agents. Each agent $i$ has local state space $\mathcal{S}_i$ and action space $\mathcal{A}_i$, and the joint spaces are $\mathcal{S} = \times_{i \in \mathcal{V} \cup \{0\}} \mathcal{S}_i$ and $\mathcal{A} = \times_{i \in \mathcal{V}} \mathcal{A}_i$, where $\mathcal{S}_0$ collects any global or unobserved state variables. At each time step $t$, agent $i$ observes $S_i^t$ and selects $A_i^t$, after which the next state is drawn from $T(\cdot | S^t, A^t)$ and a global reward $r^t = \mathcal{R}(S^t, A^t)$ is received. The setting is cooperative: all agents aim to maximise the same return. For any set of agents $N \subseteq \mathcal{V}$ and a collection of variables $\{X_i\}_{i \in \mathcal{V}}$, we write $X_N := (X_i)_{i \in N}$.

The IG $(\mathcal{V}, \mathcal{E})$ specifies which agents influence each other. We assume it is self-connected, i.e. $(i, i) \in \mathcal{E}$ for all $i$. For every agent $i$ we define the in-, out-, and undirected neighbourhoods $\overleftarrow{N}_i = \{j : (j, i) \in \mathcal{E}\}$, $\overrightarrow{N}_i = \{j : (i, j) \in \mathcal{E}\}$, and $N_i = \overleftarrow{N}_i \cup \overrightarrow{N}_i$. In a graph-based MDP (GMDP), rewards and transitions factorise according to the IG:

$$\mathcal{R}(S, A) = n^{-1} \sum_{i \in \mathcal{V}} \mathcal{R}_i(S_{\overleftarrow{N}_i}, S_0; A_{\overleftarrow{N}_i}),$$
$$T(S'|S, A) = T_0(S_0'|S_0) \prod_{i \in \mathcal{V}} T_i(S_i'|S_{\overleftarrow{N}_i}, S_0; A_{\overleftarrow{N}_i}), \quad (2.1)$$

so each agent's dynamics depend only on its local neighbourhood. Thus, in a GMDP the global reward is the average of local rewards: $r^t = \mathcal{R}(S^t, A^t) = n^{-1} \sum_{i \in \mathcal{V}} R_i^t$. Here $R_i^t := \mathcal{R}_i(S_{\overleftarrow{N}_i}^t, S_0^t; A_{\overleftarrow{N}_i}^t)$ denotes the local reward at node $i$.

A joint policy is $\pi = (\pi_1, \ldots, \pi_n)$, where each $\pi_i$ is the policy of agent $i$, mapping its local observations $\tau_i^t := (S_i^{0:t}, A_i^{0:t-1})$ to a distribution over actions in $\mathcal{A}_i$.

In the general formulation above, agents may have distinct spaces $\mathcal{S}_i$ and $\mathcal{A}_i$. In our experiments we consider homogeneous agents, i.e., $\mathcal{S}_i = \mathcal{S}_{\text{loc}}$ and $\mathcal{A}_i = \mathcal{A}_{\text{loc}}$ for all $i$, and the local transition and reward functions share the same parametric form (up to neighbourhood inputs). This symmetry allows us to share policy and critic parameters across agents, improving sample efficiency and enabling generalisation to varying graph sizes. This assumption can be relaxed by introducing type-specific policies and using a heterogeneous GNN critic (Appendix E.1).

Given discount factor $\gamma \in (0, 1)$, we define the global value function[1]

$$V^\pi(S) = \mathbb{E}_\pi \left[ \sum_{t=0}^\infty \gamma^{t+1} r^t | S^0 = S \right]. \qquad (2.2)$$

The goal is to find a joint policy $\pi^*$ that maximises $V^\pi(S)$ for all initial states $S$. For brevity, we omit the superscript $\pi$ when the policy is clear from context.

## 3. Factored value functions

While the global value function (2.2) provides the correct expected return for a joint policy, it does not decompose across agents and is impractical for large-scale MARL. Factored value functions aim to assign meaningful agent-wise value components that respect the underlying interaction structure.

### 3.1. Local value functions

In GMDPs, each agent can influence only a subset of the local rewards. Recall that $R_i^t = \mathcal{R}_i(S_{\vec{N}_i}^t, S_0^t; A_{\vec{N}_i}^t)$ denotes the reward at node $i$, and let $\vec{N}_i^m$ be the set of nodes reachable from $i$ by a directed path of length at most $m$. At time $t$, agent $i$ can influence only rewards $\{R_j^t : j \in \vec{N}_i^{t+1}\}$, motivating the *local value function*

$$V_i(S) = \mathbb{E}_\pi \left[ \sum_{t=0}^\infty n^{-1} \gamma^{t+1} \sum_{j \in \vec{N}_i^{t+1}} R_j^t \,\Big|\, S^0 = S \right]. \quad (3.1)$$

Under the GMDP factorisation (2.1), this decomposition isolates the portion of the global value that agent $i$ can directly influence (see Appendix A.1 for more details), and similar constructions have been employed in GMDP planning (Oliehoek et al., 2008; Jing et al., 2024).

However, the local value has limitations. When the average out-degree exceeds one, the neighbourhoods $N_i^{t+1}$ typically grow exponentially with $t$. As a result, local values can increase rapidly with the time horizon, producing high-variance estimates and unstable optimisation even when rewards are bounded. Appendix B.1 formalizes these limitations and Appendix A.2 shows that local values may diverge on large graphs, whereas the DVF introduced in Section 3.2 remains well-defined.

### 3.2. The diffusion value function (DVF)

In many practical environments, the influence of an agent's actions decays over both space and time. This suggests that value should propagate gradually through the interaction graph, rather than depend directly on raw rewards

from distant agents. To address the limitations identified in Appendix B.1, we introduce the *diffusion value function* (DVF), an agent-wise value construction that explicitly models this decay of influence. For additional discussion of value functions as surrogate learning objectives, see Appendix B.2. Intuitively, rewards are diffused across the graph so that nearby agents exert stronger influence than distant ones. The DVF assigns each agent a value component that incorporates both temporal discounting and spatial attenuation of rewards along the interaction graph.

Let $R^t = [R_1^t, \ldots, R_n^t]^\top \in \mathbb{R}^n$ denote the vector of agent rewards at time $t$. We define the diffusion value function as[2]

$$V_D^\pi(S) = \mathbb{E}_\pi \left[ \sum_{t=0}^\infty \Gamma^{t+1} R^t \,\Big|\, S^0 = S \right], \qquad (3.2)$$

where

$$\Gamma = \gamma \mathbb{A} D^{-1} \in \mathbb{R}^{n \times n} \quad \text{with} \quad \gamma \in (0, 1). \qquad (3.3)$$

Here, $\mathbb{A}$ is the adjacency matrix of the IG $(\mathcal{V}, \mathcal{E})$, and $D$ is the diagonal in-degree matrix with $D_{ii} = \sum_j \mathbb{A}_{ji}$.

The DVF is vector-valued, and we interpret the $i$-th component $(V_D)_i$ as the surrogate value function for agent $i$. Under this formulation, the effect of an action $A_i^t$ propagates through the influence graph according to the diffusion operator $\Gamma$, so that contributions from rewards decay smoothly with both temporal and spatial distance from $(t, i)$. The standard discount factor $\gamma$ captures temporal decay, while $\Gamma$ generalizes this by incorporating spatial decay.

This construction is most appropriate in environments where the underlying dynamics exhibit locality, i.e., when actions predominantly influence agents within short graph distances. In such settings, the DVF provides a stable and scalable alternative to the local value (3.1), particularly for infinite-horizon problems.

The operator $\Gamma$ induces a diffusion-like propagation of credit across the graph; a formal connection to random-walk diffusion is provided in Appendix B.4.

### 3.3. Theoretical properties of the DVF

The DVF satisfies a vector-valued analogue of the Bellman equation

$$V_D(S) = \Gamma \, \mathbb{E}_\pi \big[ R^t + V_D(S^{t+1}) \mid S^t = S \big], \qquad (3.4)$$

which follows directly from (3.2) for $\Gamma$ in (3.3). We first show existence (convergence of (3.2)) and uniqueness (Bellman fixed point).

---

[1]We shift the discount exponent by one for consistency with the DVF definition (3.2); this does not affect the optimisation problem.

[2]Not to be confused with the value function of (Mazoure et al., 2024), which relies on a generative model. We also distinguish *factored value functions* (as defined here using GMDP influence relations) from *factorised critic architectures* used in value decomposition methods; see Appendix B.3.

**Proposition 3.1** (Existence). *Assume rewards are bounded and $\gamma \in (0, 1)$. Then $V_D(S)$ defined in (3.2) converges and is bounded for all states $S$.*

**Proposition 3.2** (Uniqueness). *Let $\mathcal{T}^\pi$ be the Bellman operator on $V : \mathcal{S} \to \mathbb{R}^n$,*

$$(\mathcal{T}^\pi V)(S) = \Gamma \, \mathbb{E}_\pi \big[ R^t + V(S^{t+1}) \mid S^t = S \big].$$

*Since $\|\Gamma\|_1 = \max_j \sum_i \Gamma_{ij} = \gamma < 1$, $\mathcal{T}^\pi$ is a contraction on the space of bounded $V : \mathcal{S} \to \mathbb{R}^n$ under $\|V\|_\infty = \sup_S \|V(S)\|_1$, and admits a unique fixed point, which coincides with $V_D$.*

Next, we show that the DVF decomposes the global value $V$ defined in (2.2) into agent-specific components, with the DVF's average across agents recovering the global value. This ensures that $V_D$ is a faithful decomposition of the standard global value objective, enabling agent-specific learning signals while preserving $V(S)$.

**Proposition 3.3** (Factored value). *Under the GMDP reward factorisation (2.1), for any state $S$, $n^{-1} \mathbb{1}^\top V_D(S) = V(S)$.*

Finally, to connect improvements in DVF components to the global objective, we have:

**Proposition 3.4** (Policy alignment). *Let $\pi$ and $\pi'$ be two joint policies with diffusion values $V_D^\pi$ and $V_D^{\pi'}$. Under the GMDP reward factorisation (2.1), for any state $S$, if $(V_D^{\pi'}(S))_i \geq (V_D^\pi(S))_i$ for all agents $i$, with strict inequality for at least one agent, then the global value satisfies $V^{\pi'}(S) > V^\pi(S)$.*

These properties justify using the DVF for comparative policy evaluation and improvement. Although DVF-based optimisation does not in general guarantee global optimality, our empirical findings in Section 6 (Figure 2) suggest that the DVF provides a useful inductive bias for learning in graph-structured multi-agent systems. Proofs are provided in Appendix B.5.

### 3.4. Estimating the DVF

**CTDE setting.** We adopt centralised training with decentralised execution (CTDE): during execution each agent selects actions using only its local information $\tau_i$, while during training the critic may condition on global state and reward information.

**TD learning.** We learn the diffusion value function $V_D$ in (3.4) using standard temporal-difference (TD) methods (Sutton and Barto, 2018). Let $V_{D_\phi} : \mathcal{S} \to \mathbb{R}^n$ (one component per agent) denote a parametric approximation of $V_D$. Given a transition $(S^t, R^t, S^{t+1})$, we define the diffusion TD error

$$\delta^t = \Gamma \big[ R^t + V_{D_\phi}(S^{t+1}) \big] - V_{D_\phi}(S^t), \qquad (3.5)$$

and minimise $n^{-1} \|\delta^t\|^2$ using a semi-gradient update (i.e., stopping gradients through the target term).

**Decentralisability.** In our experiments, we take $V_{D_\phi}$ to be a shared GNN critic which estimates each component of the DVF by message-passing between neighbours. GNNs are known to have a bias towards short-range relations (Topping et al., 2022), which aligns well with the locality of diffusion values. We describe centralised and decentralised training schemes in Appendix E.3, but focus empirically on CTDE to avoid confounds from decentralised optimisation and communication constraints.

### 3.5. Limitations

Our approach assumes a known, time-static influence graph and a GMDP factorisation (2.1). Tasks with unknown or time-varying interaction structure, or with dense/global couplings that violate (2.1), may reduce the usefulness of DVF components and weaken our decomposition/alignment guarantees in Section 3.3. Moreover, the diffusion operator induces spatial discounting, attenuating reward information from distant agents. While this improves scalability and encourages locality in large systems, it can limit performance in domains requiring long-range coordination.

## 4. The diffusion advantage actor–critic (DA2C) algorithm

Actor–critic methods typically learn a policy (actor) together with a value function (critic) used to form low-variance policy-gradient updates. This is achieved by updating policies using an advantage estimator, often computed from a value function via TD learning.

In graph-based MDPs, the DVF provides a vector-valued critic $V_D : \mathcal{S} \to \mathbb{R}^n$ that yields agent-specific learning signals. We now introduce the diffusion advantage actor–critic (DA2C) algorithm, which integrates DVF-based advantage estimation into the standard advantage actor–critic (A2C) framework by replacing the standard scalar critic with the DVF critic $V_{D_\phi}$ and using diffusion TD errors as agent-specific advantages.

**Diffusion advantages.** Given a transition $(S^t, R^t, S^{t+1})$, let $\delta^t \in \mathbb{R}^n$ denote the diffusion TD error from (3.5). We use $\delta^t$ as a 1-step diffusion advantage estimator. More generally, a $W$-step diffusion advantage estimate is

$$\hat{G}_{D_\phi}^t = \sum_{k=0}^{W-1} \gamma^k R^{t+k} + \gamma^W V_{D_\phi}(S^{t+W}) - V_{D_\phi}(S^t), \quad (4.1)$$

obtained by unrolling (3.4) for $W$ steps and bootstrapping with $V_{D_\phi}(S^{t+W})$. For $W = 1$, (4.1) reduces to the diffusion TD error $\delta^t$. DA2C uses $(\hat{G}_{D_\phi}^t)_i$ as the advantage for

agent $i$.

**Policy update.** Under CTDE, each agent $i$ executes a policy $\pi_{\theta_i}(A_i^t \mid \tau_i^t)$ using only its local history $\tau_i^t$. DA2C updates each policy using the stochastic diffusion policy-gradient estimator

$$g_i^t = \nabla_{\theta_i} \log \pi_{\theta_i}(A_i^t \mid \tau_i^t)\,(\hat{G}_{D_\phi}^t)_i.$$

Since the DVF is a surrogate objective, $g_i^t$ is not an unbiased policy gradient estimator. However, Propositions 3.3 and 3.4 imply that the two quantities are aligned. Compared to the global (2.2) and local (3.1) value functions, the DVF is easier to estimate and typically better captures each agent's reward contribution. As a result, it tends to yield better estimates of the *counterfactual* advantage, in the sense of (Foerster et al., 2018).

DA2C alternates between updating the DVF critic by minimising $\|\delta^t\|^2$ (Section 3.4) and updating the actor using diffusion advantages. Full training details are provided in Appendix E.4.

# 5. Applications and the Learned DropEdge GNN

We evaluate DA2C on three representative graph-structured tasks: a classic cooperative MARL benchmark (Firefighting) and two distributed computation/communication problems (vector graph colouring and transmit power control). All tasks can be described or well-approximated as GMDPs under an appropriate influence graph. For the distributed computation settings, we additionally introduce the Learned DropEdge GNN (LD-GNN), a recurrent GNN architecture that learns sparse communication policies between agents.

## 5.1. Broader applicability

Graph-based MDPs arise naturally in cooperative multi-agent and networked control settings where interactions are local and can be captured by an influence graph. Examples include traffic signal control, power and energy systems, robot swarms, and network resource allocation, where rewards often decompose into local objectives and dynamics depend primarily on neighbourhood state–action variables, consistent with (2.1). DA2C provides a scalable actor–critic template for such systems by combining decentralized policies with diffusion-based value decomposition.

### 5.1.1. APPLICATION 1: FIREFIGHTING (COOPERATIVE MARL BENCHMARK)

We evaluate DA2C on a generalisation of the firefighting environment from (Oliehoek et al., 2008), where a team of firefighters cooperatively suppresses fires spreading across

a set of homes. The environment is defined on a bipartite firefighter–home graph; two firefighters influence each other if they can act on a common home, inducing a natural influence graph. At each step, firefighters move to homes to reduce fire levels, while fires spread stochastically between adjacent homes. The reward is the negative average fire level across homes. Full task details and the corresponding GMDP formulation are provided in Appendix C.1.

## 5.2. Distributed computation and communication tasks

Distributed algorithms involve a network of processors that exchange messages to achieve a common objective, with examples ranging from distributed estimation (Boyd et al., 2006; Dimakis et al., 2010) to distributed graph colouring (Barenboim and Elkin, 2022; 2009), and packet routing (Mammeri, 2019). We model such tasks as multi-agent MDPs in the LOCAL model (Linial, 1992), augmented with explicit communication costs to capture settings where message passing is constrained (e.g., due to energy or bandwidth limitations) (Baccelli et al., 2006).

**MDP formulation.** Consider a set of homogeneous, synchronized processors $\mathcal{V}$ connected by a communication graph $(\mathcal{V}, \mathcal{E})$. At time $t$, each node $i \in \mathcal{V}$ receives a local observation $O_i^t$ correlated with the global state $S^t$. Node $i$ then selects a subset of out-neighbours $C_i^t \subseteq \vec{N}_i$ to which it transmits messages, and receives messages from its in-neighbours $\{j \in \vec{N}_i : i \in C_j^t\}$. After communication, node $i$ produces a task-dependent local output $Y_i^t$ (e.g., an estimate, a colour, or a transmit-power decision). The system receives a global reward $r^t = \mathcal{R}(Y^t, C^t; S^t)$ and transitions to the next state $S^{t+1} \sim T(\cdot \mid Y^t, C^t, S^t)$. We allow the reward and transition dynamics to depend explicitly on the local outputs $Y^t$ and the communication pattern $C^t$, enabling environments where excessive communication is discouraged via a penalty term. A shared policy $\pi$ therefore specifies a distributed algorithm: it determines when and what to communicate and how to compute outputs $Y^t$ from local histories, with the goal of maximising the discounted return (2.2).

### 5.2.1. LEARNED DROPEDGE GNN (LD-GNN) FOR LEARNING DISTRIBUTED ALGORITHMS

To represent a learned distributed algorithm, the policy must decide *(i)* which messages to send and *(ii)* what local output to produce. We therefore parameterise the joint policy $\pi_\theta$ using a *message-passing policy* $\lambda_\theta$ that samples message passes $C^t$ and an *output policy* $\psi_\theta$ that produces node outputs $Y^t$. Graph neural networks (GNNs) are a natural choice, as they are built around neighbourhood message passing and align with the computation model in Section 5.2. However, standard GNN formulations typically perform message aggregation over a fixed candidate

neighbourhood (e.g., all adjacent nodes), whereas we treat communication decisions under the distributed computation constraint as a learnable component (Appendix D.1).

**LD-GNN overview.** Motivated by DropEdge (Rong et al., 2020), we propose the *Learned DropEdge GNN (LD-GNN)*, a recurrent GNN architecture that learns when to communicate by assigning locally computed edge-activation probabilities. At each time step $t$, node $i$ computes a local embedding $I_i^t$ from its observation $O_i^t$ and internal memory $X_i^t$, and samples a subset of neighbours $C_i^t \subseteq \vec{N}_i$ by independently including each $j \in \vec{N}_i$ with probability $\lambda_\theta(I_i^t, E_{ij}^t)$, where $E_{ij}^t$ denotes the local edge memory. A single-layer GNN is then applied on the induced subgraph to aggregate received messages, after which gated recurrent units (GRUs) produce the next node and edge memories, and the output policy $\psi_\theta$ samples the node output $Y_i^t$. By construction, both $C_i^t$ and $Y_i^t$ depend only on information available locally at node $i$, so LD-GNN respects the distributed computation restriction. Compared to similar GNN architectures, LD-GNN does not significantly increase computational cost at inference time, and can in fact reduce it by sparsifying the communication graph before message passing. During training, however, it incurs additional memory overhead due to the need to store edge features for time-unrolling, which is typical for recurrent GNN architectures. Figure 1 provides an overview; full architectural details are given in Appendix D.2. For distributed computation problems that admit a GMDP formulation, we train LD-GNN as the actor in DA2C, alongside a GNN diffusion-value critic $V_{D_\phi}$ (Section 3.4). This yields a flexible method that generalises across graph sizes and topologies.

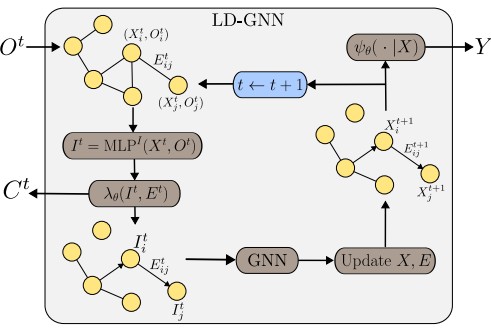

*Figure 1.* The Learned DropEdge GNN (LD-GNN) architecture. Given node and edge memory states $(X, E)$ and observations $O^t$, the policy $\lambda_\theta$ samples neighbour sets $C^t$, inducing a subgraph on which a GNN aggregates messages. GRUs update $(X, E)$, and $\psi_\theta$ samples outputs $Y^t$.

### 5.2.2. APPLICATION 2: VECTOR GRAPH COLOURING

Decentralized graph colouring is a classical distributed computation problem (Barenboim and Elkin, 2009). We study a vector-colouring variant in which each node outputs a binary colour vector and iteratively updates it based on local information and neighbour messages. The reward encourages selecting many colours while penalising conflicts with adjacent nodes, controlled by the conflict penalty $p_m$. This task admits a GMDP formulation; details are provided in Appendix C.2. We evaluate on two graph families (Appendix E.2) and show that DA2C outperforms all baselines across both topologies (Section 6 and Appendix F.2).

### 5.2.3. APPLICATION 3: TRANSMIT POWER CONTROL

We study decentralized wireless communication tasks with service-quality and energy-efficiency objectives, where rewards depend on channel capacity and interference between nearby transmitter–receiver pairs. Agents may exchange messages at a cost controlled by $p_m$, capturing energy or bandwidth constraints. A direct GMDP formulation yields a dense influence graph due to interference coupling; we therefore use a sparse communication-edge GMDP approximation that models communication decisions while preserving the reward structure. Full details of the environment and the approximate GMDP formulation are provided in Appendix C.3.1.

## 6. Experiments

To evaluate DA2C, we compare it against a range of baselines, both for cooperative graph-structured benchmarks such as the firefighting problem and for training LD-GNN policies on the distributed computation tasks introduced in Section 5, across multiple settings of the task-dependent message-passing penalty $p_m$. The role and range of $p_m$ differ by application (Appendix E.2). Overall, across all tasks and penalty settings, we evaluate over 30 distinct environment configurations.

### 6.1. Training and baselines

For each task (and penalty $p_m$ where applicable), we train and evaluate policies on independently sampled random graph instances. Graph sizes range from tens of nodes (e.g., transmit power control) to several thousand nodes (e.g., graph colouring), and OOD evaluation includes graphs with up to 10,000 nodes (Appendix F.3). Further details on graph generation and task settings are provided in Appendix E.2. Appendix E.2.5 discusses coordination-graph methods (e.g., DCG) and explains why their standard benchmark suites do not align with the GMDP setting considered here. All experiments are run on a single machine with a GTX 1080 GPU, and we report results over five independent model seeds for each algorithm and environment configuration.

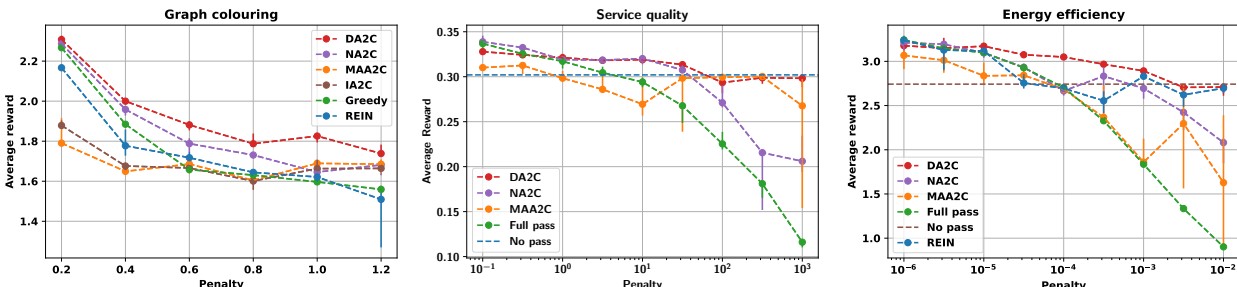

*Figure 2.* Test performance as a function of the message-passing penalty $p_m$ for the three tasks (graph colouring, service quality and energy efficiency). Error bars indicate the lower and upper quartiles across runs. We omit IA2C from the service quality task due to consistently low rewards, and from the energy efficiency task because it coincides with the no-pass baseline.

**Baselines.** To isolate the effect of the diffusion critic, we benchmark DA2C against variants of advantage actor–critic (A2C) that share the *same actor* and differ only in the critic definition. Specifically, we consider:

- **REINFORCE (REIN):** no critic network (policy gradient without baseline);

- **Independent A2C (IA2C):** per-agent critic based on local rewards,

$$V_i(S) = \mathbb{E}_\pi\Big[ \textstyle\sum_{t=0}^{\infty} \gamma^t R_i^t \,\Big|\, S^0 = S\Big];$$

- **Neighbourhood A2C (NA2C):** per-agent critic based on rewards in a fixed neighbourhood $\vec{N}_i$,

$$V_i(S) = \mathbb{E}_\pi\Big[ \textstyle\sum_{t=0}^{\infty} \gamma^t \sum_{j \in \vec{N}_i} R_j^t \,\Big|\, S^0 = S\Big];$$

- **Multi-agent A2C (MAA2C):** global critic based on the total reward,

$$V(S) = \mathbb{E}_\pi\Big[ \textstyle\sum_{t=0}^{\infty} \gamma^t r^t \,\Big|\, S^0 = S\Big].$$

All critic networks (when present) use similar GNN architectures and are trained with temporal-difference (TD) methods. Since all A2C variants share an identical actor and comparable computational cost, differences primarily reflect the effect of the critic structure. DA2C and NA2C require an additional sparse matrix–vector product in the TD update, but this overhead is negligible compared to the GNN forward pass. DA2C training and LD-GNN architecture details are provided in Appendix E.4 and Appendix E.5.

**Task-specific baselines.** For the transmit power control tasks, which include message-passing penalties, we additionally report two fixed-communication baselines: *no-pass* ($\lambda \equiv 0$) and *full-pass* ($\lambda \equiv 1$). For graph colouring, we include a classical greedy distributed baseline described in Appendix F.1.

*Table 1.* Average fire level for each method on the firefighting task (lower is better). Values show mean $\pm$ standard error over six model seeds and 100 environment instances.

| Method | Fire level |
|--------|-----------|
| DA2C | **2.637 $\pm$ 0.002** |
| NA2C | 2.713 $\pm$ 0.007 |
| MAA2C | 2.983 $\pm$ 0.006 |
| IA2C | 2.964 $\pm$ 0.003 |
| REIN | 3.01 $\pm$ 0.01 |

### 6.2. Results

Test performance is summarised in Table 1 and Figure 2, and training curves are shown in Figures 9 and 3 for the firefighting and distributed computation tasks, respectively. Across all benchmarks and (where applicable) all penalty settings, DA2C achieves the strongest and most consistent performance. In particular, the interquartile bands for DA2C are narrow in both the training curves and penalty sweeps, indicating stable learning and robust performance across penalty values $p_m$.

In contrast, MAA2C, which directly estimates the global value $V$ in (2.2), exhibits highly inconsistent performance and often underperforms the other A2C variants. This indicates that global critics provide a weak or noisy learning signal in our large-scale settings, whereas factored value functions yield more reliable agent-wise advantage estimates. Notably, DA2C remains effective even when the environment only approximately satisfies the GMDP assumptions, as in our communication tasks.

Greedy baselines such as IA2C and *no-pass* optimise each agent with respect to its own reward, ignoring the effect of its actions on neighbouring agents. As expected, these baselines perform poorly for most penalty settings, since they fail to coordinate. However, in the limit $p_m \to \infty$, communication becomes prohibitively expensive and local greedy behaviour becomes optimal; accordingly, greedy

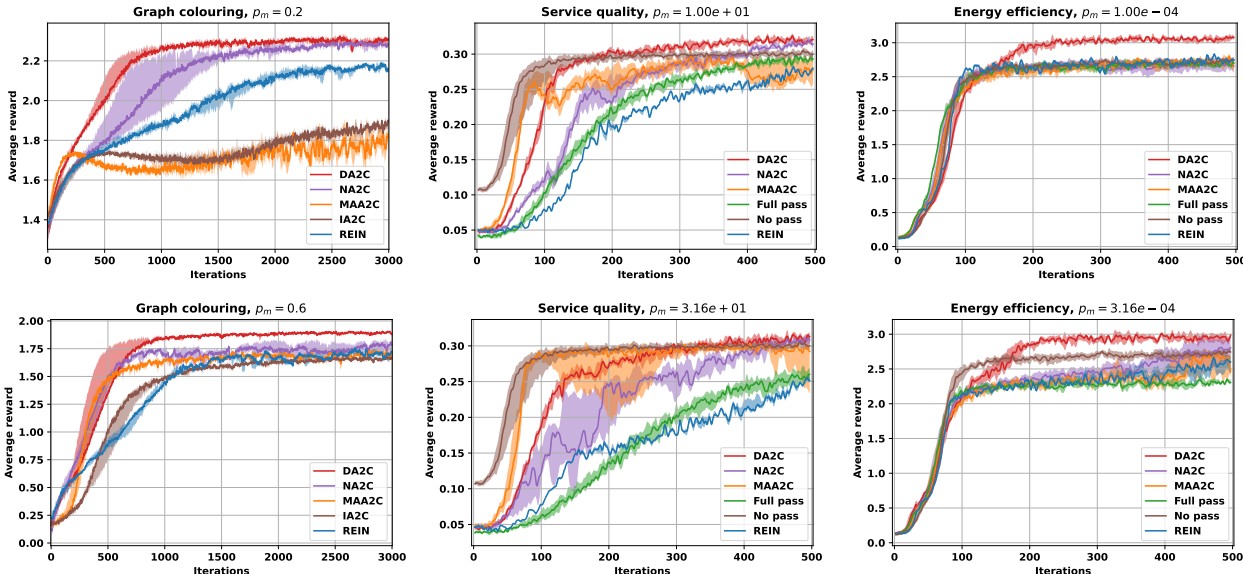

*Figure 3.* Training curves for different message-passing penalties $p_m$ on the three tasks (graph colouring, service quality and energy efficiency). Error bars indicate the lower and upper quartiles across runs. Curves are smoothed with a moving average for visual clarity.

methods approach the best achievable performance in this regime for the distributed computation tasks.

Finally, NA2C and DA2C incorporate local value signals that reflect neighbourhood influence, encouraging cooperative behaviour while remaining easier to estimate than global critics. NA2C accounts only for direct neighbours, whereas DA2C captures how influence propagates through the graph via diffusion. Both methods perform well across all tasks, with DA2C consistently outperforming NA2C, suggesting that diffusion-based credit assignment provides a favourable compromise between the scalability of local critics and the coordination capacity of global objectives.

**Ablations.** Table 2 evaluates the contribution of (i) diffusion-based value estimation and (ii) learned communication decisions in the message-penalised tasks. Replacing the DVF critic with REINFORCE (i.e., removing the critic) consistently reduces performance, indicating that diffusion-based credit assignment is important for stable learning. For the transmit-power tasks, we additionally compare against fixed-communication baselines (*full-pass/no-pass*; see Table 2 caption). Both fixed-communication baselines underperform DA2C, demonstrating the benefit of learning sparse communication under message-passing penalties. Overall, the results suggest that both the DVF critic and learned communication contribute to performance gains.

**Robustness to graph structure.** DA2C performs consistently across the graph families considered in our main experiments, including Erdős–Rényi graphs (graph colour-

*Table 2.* Ablations averaged over all message-passing penalties $p_m$ (mean reward; higher is better). For *full-pass/no-pass*, $\lambda$ is fixed and only $\psi_\theta$ is learned. N/A indicates not applicable.

| Method | Application | | |
|---|---|---|---|
| | GC | SQ | EE |
| DA2C (DVF critic) | **2.2** | **0.32** | **3.0** |
| REINFORCE (no critic) | 2.0 | 0.26 | 2.8 |
| Full-pass ($\lambda \equiv 1$) | N/A | 0.30 | 2.7 |
| No-pass ($\lambda \equiv 0$) | N/A | 0.27 | 2.4 |

ing) and random geometric graphs (transmit power), with low variability across runs (Figure 2). Appendix F.2 further evaluates DA2C on Barabási–Albert graphs, where it continues to outperform all baselines. These results suggest that DVF-based training is robust across graph topologies, while providing the largest gains in sparser settings where locality and limited influence overlap can be better exploited. In the limiting case of a fully connected graph, the DVF reduces to the standard global value function.

**Generalisation.** Appendix F.3 evaluates out-of-distribution generalisation by training on 5,000-node ER graphs and testing on larger ER graphs (10,000 nodes) and on BA graphs. DA2C remains the best-performing method across these shifts, indicating strong transfer across graph sizes and topologies.

# 7. Conclusions and future work

Our results demonstrate that the diffusion value function (DVF) is an effective critic representation for MARL on graph-based MDPs. We showed that the proposed DA2C algorithm, which leverages the DVF, achieves strong performance across over 30 environment types evaluated and often outperforms the baselines considered, yielding improvements of up to 11% in average reward.

While our experiments follow the CTDE paradigm, an important direction for future work is to develop fully decentralised DVF training schemes (see Section 3.4). In addition, our current approach assumes a fixed influence graph; extending DVF-based methods to time-varying or partially observed interaction graphs would increase their applicability to realistic large-scale systems. Finally, tackling more complex environments and richer communication constraints remains a promising avenue for further investigation.

## Acknowledgments

We gratefully acknowledge the support of the EPSRC Programme Grant EP/V026259/1, 'The mathematics of Deep Learning'.

## Impact statement

This paper advances reinforcement learning methods for graph-structured multi-agent systems, with a focus on telecommunications applications that impact network efficiency, reliability, and scalability. Potential benefits include improved communication infrastructure, reduced energy consumption, and enhanced accessibility through more efficient resource allocation. As with many optimisation methods, deployment could also introduce risks such as reinforcing inequities in service allocation, unintended interference patterns, or increased surveillance capabilities if used in network monitoring contexts. These risks depend on the deployment setting and can be mitigated through careful evaluation, constraints, and oversight in real-world systems.

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

# Appendix

# A. Properties of the local value function

## A.1. Local value as decomposition of global value

The definition of the local value function (3.1) arises naturally from the structure of the global value (2.2) for GMDPs. Recall that the global reward is $r^t = \mathcal{R}(S^t, A^t) = n^{-1} \sum_{j \in \mathcal{V}} R_j^t$ under the GMDP reward factorisation (2.1). Substituting into (2.2) gives

$$V(S) = \mathbb{E}_\pi \left[ \sum_{t=0}^\infty n^{-1} \gamma^{t+1} \sum_{j \in \mathcal{V}} R_j^t \right].$$

For any agent $i \in \mathcal{V}$, we may decompose this expression according to the out-reachable set $\vec{N}_i^{t+1}$, i.e., the set of nodes reachable from $i$ by a directed path of length at most $t + 1$:

$$V(S) = \mathbb{E}_\pi \left[ \sum_{t=0}^\infty n^{-1} \gamma^{t+1} \sum_{j \in \vec{N}_i^{t+1}} R_j^t \right] + B_i$$

$$=: V_i(S) + B_i,$$

where $B_i$ collects all reward terms associated with agents outside $\vec{N}_i^{t+1}$, i.e.,

$$B_i = \mathbb{E}_\pi \left[ \sum_{t=0}^\infty n^{-1} \gamma^{t+1} \sum_{j \in \mathcal{V} \setminus \vec{N}_i^{t+1}} R_j^t \right].$$

Crucially, $B_i$ is independent of agent $i$'s policy $\pi_i$, since under the factorisation (2.1) the influence of $\pi_i$ can propagate at most one hop per time step along the IG, and therefore cannot affect rewards outside $\vec{N}_i^{t+1}$:

**Lemma A.1** (Limited influence). *Under the GMDP factorisation* (2.1)*, for any time $t$ and any agent $j \notin \vec{N}_i^{t+1}$, the distribution of $R_j^t$ is independent of agent $i$'s policy $\pi_i$ while keeping the other agents' policies fixed.*

*Proof.* By (2.1), each local transition $T_k$ and reward $\mathcal{R}_k$ depends only on variables in $\overleftarrow{N}_k$ and the global state $S_0$. Thus agent $i$'s action can affect only its out-neighbours' next states in one step, and influence propagates at most one hop per time step along the IG. Therefore, after $t + 1$ steps, agent $i$ can affect rewards only within $\vec{N}_i^{t+1}$, implying $R_j^t$ is independent of $\pi_i$ for $j \notin \vec{N}_i^{t+1}$. $\square$

Consequently, optimising agent $i$ with respect to the local value $V_i$ yields the same policy gradient as optimising the full global value $V$. The local value therefore provides a natural decentralised objective that avoids coupling through rewards that agent $i$ cannot influence.

## A.2. Example: Unboundedness of local values on growing graphs

The standard time-discount factor $\gamma$ in MDPs does not guarantee bounded local values in GMDPs; a similar observation was made in (Oliehoek et al., 2008). We illustrate this with the following example.

Let an infinite, self-connected acyclic graph be given in which each node has degree $d > 0$. Consider a GMDP with constant rewards $r_i \equiv 1$ for all nodes $i \in \mathbb{N}_0$. The local value at node $i$ is

$$V_i = \sum_{t=0}^\infty \sum_{j \in \vec{N}_i^{t+1}} \gamma^{t+1} r_j = \sum_{t=0}^\infty \sum_{k=0}^{t+1} (d-1)^k \gamma^{t+1}.$$

This series diverges whenever $(d-1)\gamma \geqslant 1$, reflecting the exponential growth of neighbourhood size with graph distance.

In contrast, the diffusion value at node $i$ is

$$(V_D)_i = \sum_{t=0}^\infty (\mathbb{A}^{t+1})_{ii} \left( \frac{\gamma}{d} \right)^{t+1} r_i = \sum_{t=0}^\infty \gamma^{t+1},$$

which converges for all $d$ and all $\gamma \in (0, 1)$. Although our setting does not involve infinite graphs, this example illustrates that local values can grow unboundedly with graph size, whereas diffusion values remain stable.

# B. Properties of the diffusion value function

## B.1. Motivation and limitations of standard value functions

Neither the global nor the purely local value function is well suited to large-scale MARL. The global value (2.2) is conceptually aligned with the overall objective, but it becomes impractical as the number of agents grows: it fails to exploit sparsity in the interaction graph, aggregates reward noise from all agents, and admits no useful decomposition for decentralised learning.

Conversely, the local value (3.1) depends only on nearby observations, which improves scalability, but it does not reflect how influence propagates through the system. As a result, it may assign disproportionate weight to rewards generated by distant agents even when the coupling along long paths is weak. Moreover, as shown in Appendix A.2, local values can grow without bound as the graph size increases for infinite time horizons.

These limitations motivate the need for an agent-wise value construction that preserves local structure while remaining well behaved and aligned with the global objective. This motivates the DVF introduced in Section 3.2, which inter-

polates between purely local and fully global value definitions while remaining bounded and scalable.

For additional context on proposing value functions as surrogate learning objectives, see Appendix B.2.

## B.2. Designing surrogate value functions

Often, value functions are derived directly from the underlying model and objective (see (Amato and Oliehoek, 2015) for example). However, in model-free or partially specified settings it is common to optimise value functions that serve as useful surrogates, even when they are not uniquely implied by the model. A prominent example is the use of monotonic value functions in MARL (Rashid et al., 2020; Sunehag et al., 2018): although the optimal joint action-value is not guaranteed to be monotonic in environments such as StarCraft II, imposing this structure has proven effective empirically. Similarly, discounted return can be viewed as a surrogate objective: it is optimised during training even when average reward or undiscounted metrics are used for evaluation. Reward shaping techniques follow the same logic.

The local value function defined in Section 3.1 follows directly from the structure of GMDPs. The DVF, by contrast, is a surrogate construction designed to retain locality while yielding a stable and well-scaled learning signal under centralised training with decentralised execution; its local structure further suggests that decentralised training could be feasible in principle, although we do not explore this in the present work.

## B.3. Factored value functions vs. factored critic architectures

The term *factorisation* is used in different ways in the MARL literature. Popular value decomposition methods such as VDN (Sunehag et al., 2018), QMIX (Rashid et al., 2020), DCG (Boehmer et al., 2020), and DFQ (Wang and Ke, 2022) factorise the *critic network* to approximate a *global* (non-factored) value function, typically using a global reward signal. These approaches are widely studied in cooperative MARL settings often modelled as Dec-POMDPs (Oliehoek and Amato, 2016).

In contrast, *factored value functions* in the sense of GMDPs explicitly leverage the influence structure and local reward dependencies to define agent-indexed value components that can be estimated directly (Schneider et al., 1999). Such factored values have been studied in planning on structured models (Oliehoek et al., 2008), but are comparatively less explored as learnable critics in MARL. We include this note because the term "factored value function" is sometimes used ambiguously to refer to both perspectives (Naderializadeh et al., 2020; Wei et al., 2024).

## B.4. Relation between the diffusion value function and diffusion processes

The diffusion interpretation of the DVF follows from the structure of the operator $\Gamma$. The matrix $\Gamma^{\top}$ describes how credit propagates across the graph, and $\Gamma^{\top}/\gamma$ is exactly the transition matrix of a random walk on $\mathcal{G}$. In the small-scale limit such random walks converge to continuous diffusion processes, establishing a direct connection between the DVF operator and classical diffusion dynamics.

## B.5. Proofs of DVF properties

Before proving Propositions 3.1, 3.2, 3.3 and 3.4, we state the following property of $D^{-1}\mathbb{A}^{T}$:

**Lemma B.1.** *The matrix $D^{-1}\mathbb{A}^{T}$ is row-stochastic, i.e., each row sums to 1. Equivalently, $\mathbb{1}$ is a right eigenvector with eigenvalue 1: $D^{-1}\mathbb{A}^{T}\mathbb{1} = \mathbb{1}$. Further, $\mathbb{A}D^{-1}$ is column-stochastic, i.e., each column sums to 1, and hence $\mathbb{1}^{\top}\mathbb{A}D^{-1} = \mathbb{1}^{\top}$.*

*Proof.* Note that $D \in \mathbb{R}^{n \times n}$ is the (diagonal) in-degree matrix with $D_{ii} = \sum_{j} \mathbb{A}_{ji}$ and $D_{ij} = 0$ for $i \neq j$. Hence, for any row $i$, we have

$$(D^{-1}\mathbb{A}^{T})_{ik} = \frac{(\mathbb{A}^{T})_{ik}}{D_{ii}} = \frac{\mathbb{A}_{ki}}{\sum_{j} \mathbb{A}_{ji}},$$

implying that

$$\sum_{k=1}^{n} (D^{-1}\mathbb{A}^{T})_{ik} = \frac{\sum_{k} \mathbb{A}_{ki}}{\sum_{j} \mathbb{A}_{ji}} = 1.$$

Since $D$ is diagonal, $D = D^{T}$, hence

$$\mathbb{A}D^{-1} = (D^{-1}\mathbb{A}^{T})^{T}$$

and therefore the columns of $\mathbb{A}D^{-1}$ sum to 1. $\square$

*Proof of Proposition 3.1.* Recall $\Gamma = \gamma\mathbb{A}D^{-1}$ with $\gamma \in (0, 1)$. By Lemma B.1, the columns of $\mathbb{A}D^{-1}$ sum to 1. Since $\Gamma$ has nonnegative entries,

$$\|\Gamma\|_{1} = \max_{j} \sum_{i=1}^{n} \Gamma_{ij} = \gamma \max_{j} \sum_{i=1}^{n} (\mathbb{A}D^{-1})_{ij} = \gamma < 1.$$

It follows that $\|\Gamma^{k}\|_{1} \leqslant \|\Gamma\|_{1}^{k} = \gamma^{k} \to 0$ as $k \to \infty$, and the Neumann series converges:

$$\sum_{t=0}^{\infty} \Gamma^{t} = (I - \Gamma)^{-1}, \qquad \sum_{t=0}^{\infty} \Gamma^{t+1} = \Gamma(I - \Gamma)^{-1}.$$

Finally, since $R^{t}$ is bounded by assumption, the series defining $V_{D}(S)$ in (3.2) converges and $V_{D}(S)$ is bounded as well. $\square$

*Proof of Proposition 3.2.* Note that the Bellman operator $\mathcal{T}^\pi$ on vector-valued functions $V : \mathcal{S} \to \mathbb{R}^n$ is given by

$$(\mathcal{T}^\pi V)(S) = \Gamma \, \mathbb{E}_\pi\big[R^t + V(S^{t+1}) \mid S^t = S\big].$$

Recall that $\Gamma = \gamma \mathbb{A} D^{-1}$ with $\gamma \in (0,1)$. By Lemma B.1, the columns of $\mathbb{A} D^{-1}$ sum to 1. Since $\Gamma$ has nonnegative entries, it follows that

$$\|\Gamma\|_1 = \max_j \sum_i \Gamma_{ij} = \gamma < 1.$$

Now consider the norm $\|V\|_\infty := \sup_{S \in \mathcal{S}} \|V(S)\|_1$. For any two functions $V, W$ and any state $S$,

$$\begin{aligned}
&\|(\mathcal{T}^\pi V)(S) - (\mathcal{T}^\pi W)(S)\|_1 \\
&= \big\|\Gamma \, \mathbb{E}_\pi\big[V(S^{t+1}) - W(S^{t+1}) \mid S^t = S\big]\big\|_1 \\
&\leqslant \|\Gamma\|_1 \, \mathbb{E}_\pi\big[\|V(S^{t+1}) - W(S^{t+1})\|_1 \mid S^t = S\big] \\
&\leqslant \|\Gamma\|_1 \|V - W\|_\infty.
\end{aligned}$$

Taking the supremum over $S$ yields

$$\|\mathcal{T}^\pi V - \mathcal{T}^\pi W\|_\infty \leqslant \|\Gamma\|_1 \|V - W\|_\infty = \gamma \|V - W\|_\infty.$$

Thus $\mathcal{T}^\pi$ is a contraction on the space of bounded $V : \mathcal{S} \to \mathbb{R}^n$ under $\|V\|_\infty = \sup_S \|V(S)\|_1$. By the Banach fixed-point theorem, it admits a unique fixed point, which coincides with the diffusion value function $V_D$ by (3.4). $\quad\square$

*Proof of Proposition 3.3.* Let $\mathbb{1}$ denote the $n$-dimensional all-ones vector. Using the definition of the DVF in (3.2) and $\Gamma = \gamma \mathbb{A} D^{-1}$ with $\gamma \in (0,1)$ we can compute the element-wise average as

$$\begin{aligned}
&n^{-1} \mathbb{1}^\top V_D(S) \\
&= \mathbb{E}\left[\sum_{t=0}^\infty n^{-1} \mathbb{1}^\top \Gamma^{t+1} R^t \mid S^0 = S\right] \\
&= \mathbb{E}\left[\sum_{t=0}^\infty n^{-1} \gamma^{t+1} \mathbb{1}^\top (\mathbb{A} D^{-1})^{t+1} R^t \mid S^0 = S\right].
\end{aligned}$$

Lemma B.1 ensures $\mathbb{1}^\top \mathbb{A} D^{-1} = \mathbb{1}^\top$, implying $\mathbb{1}^\top (\mathbb{A} D^{-1})^{t+1} = \mathbb{1}^\top$ for all $t \geqslant 0$ and hence

$$n^{-1} \mathbb{1}^\top V_D(S) = \mathbb{E}\left[\sum_{t=0}^\infty n^{-1} \gamma^{t+1} \mathbb{1}^\top R^t \mid S^0 = S\right].$$

Under the GMDP reward factorisation (2.1),

$$r^t = n^{-1} \mathbb{1}^\top R^t,$$

and therefore

$$n^{-1} \mathbb{1}^\top V_D(S) = \mathbb{E}\left[\sum_{t=0}^\infty \gamma^{t+1} r^t \mid S^0 = S\right] = V(S),$$

i.e., the average diffusion value equals the global value. $\quad\square$

*Proof of Proposition 3.4.* Let $\mathbb{1}$ denote the $n$-dimensional all-ones vector. By Proposition 3.3,

$$V^\pi(S) = n^{-1} \mathbb{1}^\top V_D^\pi(S)$$

under the GMDP reward factorisation (2.1) and similarly for $V^{\pi'}(S)$. Hence,

$$V^{\pi'}(S) = n^{-1} \mathbb{1}^\top V_D^{\pi'}(S) > n^{-1} \mathbb{1}^\top V_D^\pi(S) = V^\pi(S),$$

since at least one component of $V_D^{\pi'}(S)$ is strictly larger. $\quad\square$

## C. Task formulations

We provide more details on our applications: the firefighting application in Section C.1, the vector graph colour application in Section C.2 and two transmit power control applications (service quality and energy efficiency rewards) in Section C.3.

### C.1. Firefighting application

We formulate the firefighting environment of (Oliehoek et al., 2008) as a GMDP consistent with (2.1). The environment is defined on a bipartite graph between firefighters $\mathcal{V}$ and homes $\mathcal{H}$. Each home $h \in \mathcal{H}$ has a fire level $f_h^t \in \{0, \dots, f_{\max}\}$ and is initialised with $f_h^0$ sampled uniformly from $\{0, \dots, f_{\max}\}$. A home $h$ is burning if $f_h^t > 0$.

At each time step, the fire level of a home increases by one with probability $0.8$ if at least one adjacent home is burning (where home adjacency is induced by sharing a firefighter neighbour); otherwise, if the home itself is burning, its fire level increases with probability $0.4$. Each firefighter $i \in \mathcal{V}$ selects a home $h \in N_i$ to move to, where $N_i \subseteq \mathcal{H}$ denotes the set of homes adjacent to firefighter $i$ in the bipartite graph. If exactly one firefighter moves to a home, its fire level decreases by one; if two or more move to the same home, the fire is extinguished ($f_h^t = 0$).

**Global reward.** We define the global reward as the negative average fire level across homes,

$$r^t = |\mathcal{H}|^{-1} \sum_{h \in \mathcal{H}} -f_h^t.$$

**Influence graph and local rewards.** We define the influence graph over firefighters by

$$(i,j) \in \mathcal{E} \iff N_i \cap N_j \neq \varnothing,$$

i.e., two firefighters influence each other if they can act on a common home. This graph is self-connected provided each firefighter is adjacent to at least one home.

To obtain a reward factorisation consistent with (2.1), we define a local reward for each firefighter $i$ by distributing

each home's contribution equally among its adjacent firefighters:

$$R_i^t = \sum_{h \in N_i} -\frac{f_h^t}{|N_h|}.$$

Here, $N_h := \{i \in \mathcal{V} : h \in N_i\}$ is the set of firefighters adjacent to home $h$ and we assume each home is adjacent to at least one firefighter, so $|N_h| \geqslant 1$. Then

$$\sum_{i \in \mathcal{V}} R_i^t = \sum_{i \in \mathcal{V}} \sum_{h \in N_i} -\frac{f_h^t}{|N_h|} = \sum_{h \in \mathcal{H}} \sum_{i \in N_h} -\frac{f_h^t}{|N_h|} = \sum_{h \in \mathcal{H}} -f_h^t.$$

Thus $r^t = |\mathcal{H}|^{-1} \sum_{i \in \mathcal{V}} R_i^t$; equivalently, defining $\tilde{R}_i^t := \frac{|\mathcal{V}|}{|\mathcal{H}|} R_i^t$ yields

$$r^t = |\mathcal{V}|^{-1} \sum_{i \in \mathcal{V}} \tilde{R}_i^t,$$

matching (2.1) exactly.

**Local transition dependence.** The fire dynamics of a home $h$ depend only on (i) the current fire levels of $h$ and its adjacent homes, and (ii) the set of firefighters that move to $h$ at time $t$. The latter depends only on the actions of firefighters in $N_h$. Consequently, the evolution of the fire levels in the neighbourhood of firefighter $i$ depends only on the actions of firefighters $j$ with $N_i \cap N_j \neq \varnothing$, i.e., on its neighbours in the influence graph. Therefore, rewards and transitions are local with respect to this graph, yielding a valid GMDP formulation.

## C.2. Vector graph colouring application

In this section, we provide a mathematical formulation of the vector graph colouring application. Let $c \in \mathbb{N}$ be the number of colours available. With reference to Section 5.2, we consider an undirected communication graph $(\mathcal{V}, \mathcal{E})$. At time $t$, each node $i \in \mathcal{V}$ outputs a binary vector $Y_i^t \in \{0, 1\}^c$ which determines its colour assignment. We define the reward $r^t = |\mathcal{V}|^{-1} \sum_{i \in \mathcal{V}} R_i^t$, where

$$R_i^t = \langle Y_i^t, Y_i^t \rangle - p_m \sum_{j \in N_i} \langle Y_i^t, Y_j^t \rangle.$$

This reward encourages each node to allocate as many colours as possible while avoiding the colours allocated by neighbours, with the parameter $p_m$ determining the penalty for colour conflicts.

Randomised algorithms are known to be more efficient than deterministic ones for distributed graph colouring (Barenboim et al., 2016) due to graph symmetries which need to be broken if a good colouring is to be found. To introduce randomisation, we give each node a random number as its observation $O_i \sim U(0, 1)$. This is a minor change, but we found that it yields significant improvements in performance.

Since message passing is not penalised, we assume that messages are always passed between neighbouring agents. We can describe this task as a GMDP where the influence and communication graphs are identical. The task admits an agent-wise reward factorisation via $R_i^t$ and has no hidden state $S_0 = \varnothing$. Each local state $S_i^t$ consists of the information available at node $i$ at time $t$ (e.g., its local observation and the most recent neighbour messages). Local states thus represent the information used by each agent to compute its output $Y_i^t$.

## C.3. Transmit power control applications

We consider two multi-agent wireless network tasks, namely service quality maximisation and energy efficiency optimisation, and present a formal model for each, along with an approximate GMDP formulation suitable for MARL.

C.3.1. NETWORK AND OBSERVATION MODEL

We model a wireless network as a set of linked transmitter-receiver (TX-RX) pairs forming a weighted, undirected communication graph $(\mathcal{V}, H)$, shown in Figure 4. Each node $i \in \mathcal{V}$ represents a TX-RX pair and the edge weight $H_{ij} = H_{ji} \in [0, 1]$ encodes the proportion of power transmitted at node $i$ that interferes with the receiver at node $j$ (further details in Appendix E.2).

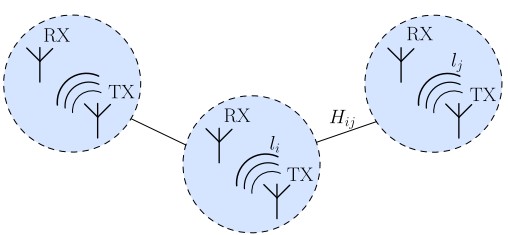

*Figure 4.* TX-RX communication graph.

At each time $t$, node $i$ observes

$$O_i^t = (\alpha_i^t, \beta_i^t, l_i^t, \mathcal{I}_i^{t-1}),$$

where $\alpha_i^t, \beta_i^t$ are user-specific service quality parameters, $l_i^t \in [0, 1]$ is the received-to-transmitted power ratio, and $\mathcal{I}_i^t$ is the interference at the receiver. The variables $\alpha_i^t, \beta_i^t, l_i^t$ evolve as bounded Gaussian random walks (BGRW), ensuring values remain within specified ranges. For use in neural networks, BGRWs can be normalised (see Appendix C.3.4).

The interference at node $i$ depends on the power transmitted by its neighbouring nodes, and is given by

$$\mathcal{I}_i^t = \sum_{j \in N_i} H_{ji} \, p_j^t + N_0,$$

where $p_j^t$ is the total transmit power of node $j$, $N_0$ is the noise floor, and $N_i$ is the set of neighbours of $i$. The total transmit power at node $i$ is

$$p_i^t = Y_i^t + p_m|C_i^t| + p_0,$$

where $Y_i^t$ is the controllable transmit power (the agent's action), $|C_i^t|$ counts the messages passed by node $i$, and $p_m, p_0$ are constants representing message-passing and baseline power consumption, respectively. This linear model reflects per-message transmission and processing costs and is commonly used as a first-order approximation in communication-aware MARL.

For notational convenience, we represent the received interference explicitly as $\mathcal{I}_i^t$ rather than implicitly through the channel matrix $H$. This allows us to write the interference vector compactly as $\mathcal{I}^t = Hp^t + N_0\mathbb{1}$, where $p^t$ is the transmit-power vector and $\mathbb{1}$ is the all-ones vector. Explicitly separating $\mathcal{I}_i^t$ also makes it easy to isolate the desired-signal term from aggregate interference and noise when computing capacity, and matches the vectorised implementation used in our code.

### C.3.2. REWARDS

In this subsection, we introduce the rewards for the two tasks: service quality and energy efficiency.

**Service quality.** Service quality is often measured as the data rate achieved by a transmitter, but the relationship between data rate and perceived quality is typically non-linear and user-dependent. For example, inelastic traffic such as phone calls experiences diminishing returns at high rates (Pham and Hwang, 2016). We model this behavior using a user-specific sigmoid function

$$q_{\alpha,\beta}(c) = \frac{1 - \exp(-c/\beta)}{1 + \exp((\alpha - c)/\beta)} \in [0, 1),$$

where $c$ is the channel capacity, and $\alpha, \beta$ control the steepness and range of quality increase. This functional form captures saturation effects while remaining smooth and bounded, which stabilises policy optimisation. For node $i$ at time $t$, the channel capacity is $c_i^t = \log_2\left(1 + l_i^t Y_i^t / \mathcal{I}_i^t\right)$ and the corresponding service quality is $q_{\alpha_i^t, \beta_i^t}(c_i^t)$. This yields the global reward

$$r_{\text{SQ}}^t = \frac{1}{|\mathcal{V}|} \sum_{i \in \mathcal{V}} q_{\alpha_i^t, \beta_i^t}(c_i^t).$$

**Energy efficiency.** Energy efficiency balances service quality against power consumption. For node $i$ at time $t$, we define

$$\varphi_i^t = \frac{q_{\alpha_i^t, \beta_i^t}(c_i^t)}{p_i^t},$$

resulting in the global rewards

$$r_{\text{EE}}^t = \frac{1}{|\mathcal{V}|} \sum_{i \in \mathcal{V}} \varphi_i^t,$$

penalising excessive message passing through the dependence on $p_m$.

### C.3.3. MODEL ASSUMPTIONS

We note that in centralised training, there is no need to directly measure the channel capacity $c_i^t$ for each node, since it can be computed centrally using a known model of the environment. Training is performed offline in simulation rather than on the physical system. In real communication systems, interference can be estimated by measuring the total received signal power during intervals when the transmitter is silent. Measuring $c_i^t$ (e.g., to select modulation and coding schemes) is typically done using pilot signals. Our model follows this common structure: precise during centralised training, while relying on local estimation mechanisms during execution.

In our setup, the exchanged messages are essentially control signals carrying minimal information. For simplicity, we assume they are delivered reliably, which is a common abstraction in multi-agent communication settings. Packet loss or rate constraints could be incorporated by randomly dropping edges or introducing message noise; this is left as a direction for future extensions. Moreover, we assume that data transmission and message exchange occur simultaneously. In a real communication system, these would typically be multiplexed to avoid interference, but we adopt this simplification to maintain a structured MDP that captures key challenges such as interference and coordination, while keeping the environment tractable for evaluation.

### C.3.4. BOUNDED GAUSSIAN RANDOM WALKS (BGRW)

A BGRW is a sequence $x_1, x_2, x_3, \ldots$ where $x_1$ is chosen uniformly from an interval $[x_{\min}, x_{\max}]$ and $x_t = \text{Clip}(y_t, x_{\min}, x_{\max})$ for $t > 1$, where $y_t \sim N(x_{t-1}, \sigma^2)$ and the clip function Clip is defined as

$$\text{Clip}(y_t, x_{\min}, x_{\max}) = \begin{cases} x_{\min}, & y \leqslant x_{\min}, \\ y, & y \in (x_{\min}, x_{\max}), \\ x_{\max}, & y \geqslant x_{\max}. \end{cases}$$

Since $p_0 > 0$, the denominator is strictly positive. When used as input to an ML model, a BGRW can be normalised as

$$\bar{x} = \frac{2x - x_{\max} - x_{\min}}{4\sqrt{3}(x_{\max} - x_{\min})}.$$

Here, the scaling constant $4\sqrt{3}$ normalises the variance of a uniform random variable on $[x_{\min}, x_{\max}]$ to approximately unit variance, improving numerical stability.

### C.3.5. APPROXIMATE GMDP FORMULATION

We treat transmit-power decisions $Y$ as part of the environment dynamics and focus learning on the communication policy, which determines which messages are exchanged. We formulate each communication edge $(i, j)$ as a GMDP agent that decides whether to transmit a message from $i$ to $j$. Using an exact GMDP description for the transmit power tasks results in a very dense influence graph, which is computationally impractical. We therefore employ a sparser approximation that focuses on message passing dynamics while keeping the node outputs $Y$ fixed. This formulation provides the reward and observation structures necessary to applying the diffusion value function (DVF) and GNN-based MARL methods described in Section 5.2.1.

In this formulation, the passing policy acts independently on each edge. Consequently, each communication edge $(i, j) \in \mathcal{E}$ can be treated as a GMDP agent with two discrete actions: either transmit a message from node $i$ to $j$, or do not transmit. This yields an *edge GMDP* defined on the influence graph $(\mathcal{E}, \mathcal{L})$, illustrated in Figure 5.

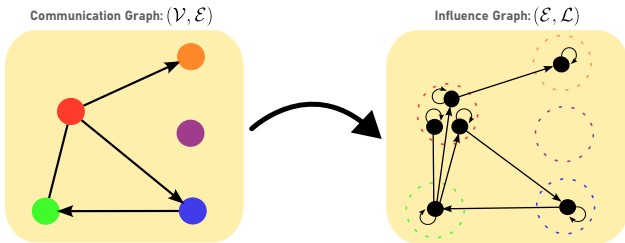

*Figure 5.* Transformation from a communication graph to the influence graph of its edge GMDP.

To define edge rewards, we first compute smoothed node rewards

$$U^t = \mathbb{A} D^{-1} R^t, \qquad (C.1)$$

and assign to each edge agent $(i, j)$ a reward

$$R_{(i,j)}^t = \frac{U_i^t}{|N_i|}.$$

This ensures that the sum of edge rewards equals the sum of node rewards, i.e.,

$$\sum_{e \in \mathcal{E}} R_e^t = \sum_{i \in \mathcal{V}} R_i^t,$$

so the edge GMDP preserves the proportionality of rewards ensuring the edge GMDP yields a communication policy that is consistent with the original objective.

Passing a message $(i, j)$ has three effects: (i) it delivers information to node $j$, (ii) consumes energy at node $i$, and (iii) generates interference at nodes in $N_i$. Thus, edge $(i, j)$ influences all nodes in $N_i \cup \{i\}$, and consequently all edge agents $(k, l)$ with $k \in N_i \cup \{i\}$. For a communication graph

with average degree $d$, this would yield an edge influence graph $(\mathcal{E}, \mathcal{L})$ with average degree $\mathcal{O}(d^2)$ too dense for practical learning.

To reduce complexity, we focus only on information transmission as the source of influence. The resulting edge influence graph is

$$\mathcal{L} = \{(e, f) \in \mathcal{E}^2 : e = f \text{ or } e_1 = f_0\},$$

where $e_0$ and $e_1$ denote the start and end nodes of edge $e$. In other words, $\mathcal{L}$ contains self-connections and directed edges linking edges whose head coincides with another edge's tail, while preserving directionality.

Although energy consumption and interference effects are ignored in the influence graph, the reward averaging in (C.1) ensures that these factors still contribute to edge rewards.

Further implementation details, including the actor network architecture for learning the edge GMDP policy, are provided in Appendix E.5.

## D. Learned DropEdge GNN (LD-GNN) details

### D.1. Message passing as communication decisions

Standard GNN formulations typically perform message aggregation over a fixed candidate neighbourhood (e.g., all adjacent nodes), learning message functions and/or attention weights (Velickovic et al., 2017) within that set. In contrast, our setting treats *communication decisions* (i.e., which node pairs exchange messages) as a learnable component under the distributed computation constraint in Section 5.2. Specifically, we learn which communication links to use under distributed-system constraints, treating sparse communication patterns as an explicit decision variable. Concretely, the message-passing policy $\lambda_\theta$ learns a distribution over message passes $C^t$, inducing a sparse communication graph at each iteration. Unlike attention weights, which define continuous importance scores over candidate edges, $C^t$ represents an explicit (potentially sparse) communication decision that determines which messages are actually sent. This distinction is crucial when communication is costly.

Related work explores learned sparsification or restricted message passing (Luo et al., 2021; Xiao et al., 2021); however, these approaches are not designed to enforce the distributed computation constraint in Section 5.2.

### D.2. Architecture details

This subsection provides full details of the Learned DropEdge GNN (LD-GNN) actor architecture introduced in Section 5.2.1. LD-GNN implements a learned distributed al-

gorithm by jointly learning *(i)* which messages to send and *(ii)* what local output to produce, using only locally available information at each node.

**Node and edge memories.** For each node $i \in \mathcal{V}$, LD-GNN maintains a node memory state $X_i^t \in \mathbb{R}^{d_x}$. For each directed edge $(i, j) \in \mathcal{E}$, we additionally maintain an edge memory state $E_{ij}^t \in \mathbb{R}^{d_e}$ encoding information about communication from $i$ to $j$. These memory states allow LD-GNN to implement multi-round distributed computation via recurrence.

**Local embeddings.** At time step $t$, node $i$ receives an observation $O_i^t$ and computes a local embedding

$$I_i^t = \mathrm{MLP}_\theta^I(X_i^t, O_i^t),$$

which summarises the node's current information and internal memory.

**Sampling message passes.** LD-GNN learns a message-passing policy $\lambda_\theta$ which assigns, independently for each directed edge $(i, j)$, a probability of sending a message from node $i$ to node $j$:

$$\Pr(j \in C_i^t) = \lambda_\theta(I_i^t, E_{ij}^t) \in (0, 1),$$

where $C_i^t \subseteq \vec{N}_i$ denotes the set of neighbours that node $i$ chooses to communicate with at time $t$. The resulting communication sets $C^t = \{C_i^t\}_{i \in \mathcal{V}}$ induce an active edge set

$$\mathcal{F}^t = \{(i, j) \in \mathcal{E} : j \in C_i^t\},$$

which defines the subgraph on which message passing is performed.

**Message aggregation with a GNN.** Given the induced subgraph $\mathcal{F}^t$ and the node embeddings $I^t = \{I_i^t\}_{i \in \mathcal{V}}$, LD-GNN computes a message-aggregated representation

$$Z^t = \mathrm{GNN}_\theta(I^t; \mathcal{F}^t),$$

where $\mathrm{GNN}_\theta$ is a single message-passing layer applied on the active edges only. In our implementation, $Z_i^t$ aggregates information from incoming neighbours $j$ such that $(j, i) \in \mathcal{F}^t$.

**Memory updates.** The node memory states are updated using a gated recurrent unit:

$$X_i^{t+1} = \mathrm{GRU}_\theta^X(Z_i^t; X_i^t).$$

Edge memories are updated only for edges that successfully transmitted a message. Specifically, for each directed edge $(i, j) \in \mathcal{E}$,

$$E_{ij}^{t+1} = \begin{cases} \mathrm{GRU}_\theta^E(I_i^t, I_j^t; E_{ij}^t), & (i, j) \in \mathcal{F}^t, \\ E_{ij}^t, & (i, j) \notin \mathcal{F}^t. \end{cases}$$

This update allows each edge memory to accumulate information about past communication events.

**Output policy.** Finally, each node samples its task-dependent output using an output policy $\psi_\theta$ conditioned on the updated node memory:

$$Y_i^t \sim \psi_\theta(\cdot \mid X_i^{t+1}).$$

The output space depends on the application: for example, in graph colouring $Y_i^t \in \{0, 1\}^c$, while in other tasks it may be discrete or continuous.

**Distributed computation restriction.** All computations at node $i$ are functions only of $O_i^t$, the node memory $X_i^t$, and edge memories $\{E_{ij}^t : j \in \vec{N}_i\}$. Thus, both the message-passing decisions $C_i^t$ and the output $Y_i^t$ are generated using only locally available information, ensuring that LD-GNN satisfies the distributed computation restriction of Section 5.2.

**Special cases.** If communication is always permitted and not penalised (e.g. graph colouring), we set $\lambda_\theta \equiv 1$, so that $\mathcal{F}^t = \mathcal{E}$ for all $t$. In this case LD-GNN reduces to a recurrent GNN actor which learns only the output policy $\psi_\theta$. Task-specific implementation details are provided in Appendix E.5.

# E. Implementation details

This appendix provides additional implementation details. We describe the extension to heterogeneous agents in Section E.1, the generation of the influence graphs used in our synthetic environments in Section E.2, and the DVF training schemes in Section E.3. Implementation details for the DA2C algorithm are given in Section E.4. Finally, we describe the LD-GNN architecture in Section E.5, which is trained using DA2C.

### E.1. Extension to heterogeneous agents

Our theoretical framework does not require homogeneous agents. In settings with multiple agent types $c(i) \in \{1, \ldots, K\}$, one can use type-specific policies $\pi_{\theta_{c(i)}}$ and critics $V_{\phi_{c(i)}}$. For centralised DVF estimation, a heterogeneous GNN can be used to compute $V_D(S)$ by conditioning message passing on agent types. The distributed DVF training scheme remains unchanged, with each agent maintaining a local critic and exchanging the same neighbour tuples.

### E.2. Environment generation

This section describes how we generate influence graphs for the synthetic test environments. We use synthetic environments because widely used MARL benchmarks such

as StarCraft II (De Witt et al., 2020) do not expose a clear GMDP structure with explicit local rewards and influence graphs, and they are relatively small compared to the large-scale settings targeted by our approach. Moreover, we are not aware of standard MARL baselines tailored to distributed computation at the scales considered here.

### E.2.1. GRAPH DISTRIBUTIONS AND MOTIVATION

We evaluate our method across three families of graph topologies to test robustness across distinct structural regimes: Erdős–Rényi (ER) graphs (Erdős and Rényi, 1959), Barabási–Albert (BA) graphs (Albert and Barabási, 2002), and random geometric graphs. These distributions capture qualitatively different network structures and are widely used models of networked interactions. ER graphs are commonly used as standard random graph benchmarks (e.g., (Erdős and Rényi, 1959; Hu et al., 2023)). They typically model relatively homogeneous interaction networks (e.g., ad hoc or sensor networks), where degree distributions concentrate and clustering is limited. In contrast, BA graphs exhibit heavy-tailed degree distributions with hubs and pronounced clustering, often used to model scale-free networks such as citation and internet graphs. Random geometric graphs are widely used for wireless connectivity modeling (e.g., (Eisen and Ribeiro, 2020)) and are a natural choice for our transmit power control tasks.

### E.2.2. FIREFIGHTING

For the firefighting task, we considered environments with 10,000 homes and 5,000 firefighters. The connections between homes and firefighters sampled as an ER graph with two constraints: each firefighter is connected to at least two homes, and each home is connected to at least one firefighter. This ensures no trivial topologies are sampled.

### E.2.3. GRAPH COLOURING: ER AND BA GRAPHS

For graph colouring, we evaluate two random graph families: ER graphs (Erdős and Rényi, 1959) with $n = 5,000$ nodes and mean degree 3 (Section 6), and BA graphs (Albert and Barabási, 2002) with $n = 1,000$ nodes and attachment parameter $m = 3$ (Section F.2).

We also note a practical difference in generation cost: ER graphs can be generated efficiently in a parallel fashion, enabling scaling to thousands of nodes with minimal overhead. In contrast, BA graphs require sequential construction due to preferential attachment, making large instances substantially more expensive to generate; we therefore limit BA graphs to 1,000 nodes.

### E.2.4. TRANSMIT POWER TASKS: RANDOM GEOMETRIC GRAPHS

For the transmit power control environments, we construct random geometric graphs, which are standard models for wireless connectivity (e.g., (Eisen and Ribeiro, 2020)). We first sample the number of nodes $n$ uniformly from a range $[n_{\min}, n_{\max}]$. We then draw node positions $p_i$ i.i.d. uniformly from the unit square $[0, 1]^2$, and add an undirected edge $(i, j) \in \mathcal{E}$ if $\|p_i - p_j\|_2 < d$ for a fixed threshold $d > 0$. For the service quality task we use $n_{\min} = 20, n_{\max} = 50$, and for the energy efficiency task we use $n_{\min} = 10, n_{\max} = 50$. When interference channels are required, we compute them as $H_{ij} = (\|p_i - p_j\|_2 + 0.1)^{-5}$.

.

### E.2.5. COMPARISON TO COORDINATION-GRAPH METHODS (E.G., DCG)

Deep Coordination Graphs (DCG) (Boehmer et al., 2020) and related value factorisation methods are typically evaluated in general cooperative MARL benchmarks, where an explicit influence graph and local reward factorisation are not provided. DCG factorises a global value function into pairwise terms, and must therefore infer or approximate interaction structure from experience. In contrast, our work focuses on GMDPs, where the influence structure is part of the model and induces an agent-wise value construction that aggregates effects over graph neighbourhoods, which may extend beyond pairwise interactions. In our experiments, we also include MAA2C, which estimates a global critic from local information and provides a complementary point of comparison to DCG-style methods.

Extending DA2C to the benchmark suites commonly used for DCG-style methods is an interesting direction for future work. This would require learning or approximating the underlying influence structure in settings where it is not explicitly specified, enabling DVF-based critics to be applied in more general cooperative MARL environments. In this paper, we focus on rigorous development and evaluation in domains where the influence structure is explicit and well-defined.

### E.3. DVF training schemes

In this section, we describe practical methods for estimating the DVF using both centralised and distributed training paradigms, corresponding to the training regimes outlined in Section 3.4.

### E.3.1. CENTRALISED TRAINING

In centralised training, global state and reward information is available, and parameter sharing across agents can be used (Zhang et al., 2021). Actor–critic methods typically

employ either a global critic using the full state or local critics based on each agents observation history $\tau_i$ (Li et al., 2022).

For MARL problems on an IG, it is natural to estimate $(V_D(S))_i$ from the observation histories within an $m$-hop undirected neighbourhood $N_i^m$. This encourages cooperation among nearby agents while ignoring redundant distant information. Under homogeneous agents, an approximation $V_\phi(\tau_{N_i^m}) \approx (V_D(S))_i$ allows parameter sharing, enabling us to estimate the DVF for all agents using a graph neural network (GNN), i.e., $\text{GNN}_\phi(\tau; \tilde{\mathbb{A}}) \approx V_D(S)$, where $\tilde{\mathbb{A}}_{ij} = \max\{\mathbb{A}_{ij}, \mathbb{A}_{ji}\}$ is the symmetrised IG. The number of GNN message-passing layers determines the critic's receptive field, allowing it to interpolate smoothly between local and global critics (Kortvelesy and Prorok, 2022; Nayak et al., 2023). This aligns well with the DVF, whose contributions decay with graph distance. GNNs therefore provide a natural architecture for DVF estimation in environments with arbitrary agent counts, as they effectively capture short-range dependencies while avoiding the difficulties associated with long-range credit assignment (Topping et al., 2022).

### E.3.2. DISTRIBUTED TRAINING

For the distributed DVF estimation scheme, each agent $i$ maintains local parameters $\phi_i$ and learns a scalar estimate of its DVF component, $V_{\phi_i}(\tau_i^t) \approx (V_D(S^t))_i$, from local information $\tau_i^t$ and neighbour messages.

Using $\Gamma = \gamma \mathbb{A} D^{-1}$, agent $i$ forms the local Bellman target

$$y_i^t := \big(\Gamma[R^t + V_\phi(\tau^{t+1})]\big)_i = \sum_{j \in \vec{N}_i} \frac{\gamma \mathbb{A}_{ij}}{d_j}\Big(R_j^t + V_{\phi_j}(\tau_j^{t+1})\Big),$$

$$(E.1)$$

where $\vec{N}_i$ is the out-neighbourhood of agent $i$ and $d_j = D_{jj} = \sum_k \mathbb{A}_{kj}$ is the in-degree of $j$ used in $\Gamma$. Agent $i$'s local TD error is the $i$-th component of the global vector TD error in Section 3.4,

$$\delta_i^t = (\delta^t)_i = y_i^t - V_{\phi_i}(\tau_i^t).$$

Minimising the global TD loss $n^{-1}\|\delta^t\|^2 = n^{-1}\sum_{i=1}^n (\delta_i^t)^2$ therefore decomposes into per-agent contributions $n^{-1}(\delta_i^t)^2$, which can be optimised independently by each agent.

To compute $y_i^t$, agent $i$ only requires the tuple $\{(R_j^t, V_{\phi_j}(\tau_j^{t+1}), d_j)\}_{j \in \vec{N}_i}$ from its neighbours. These quantities can be exchanged via local message passing, enabling fully decentralised critic learning without a central coordinator. The per-step communication cost is $O(|\vec{N}_i|)$.

### E.4. DA2C training

We provide more details on the DA2C algorithm introduced in Section 4. DA2C alternates between updating a DVF critic $V_{D_\phi}$ via TD learning and updating the policy parameters $\theta$ using diffusion advantages. We consider $N$ policy iterations of length $M$.

**Diffusion advantages.** We compute diffusion advantages $\hat{G}_{D_\phi}^t \in \mathbb{R}^n$ using the $W$-step estimator (4.1) and use $(\hat{G}_{D_\phi}^t)_i$ as the advantage signal for agent $i$.

**Critic update.** We write $\text{sg}(\cdot)$ to denote the stop-gradient operator, i.e., $\text{sg}(x) = x$ in the forward pass and $\nabla \text{sg}(x) = 0$. The critic parameters $\phi$ are trained using semi-gradient steps which minimise the TD loss $n^{-1}\|\delta^t\|^2$, where

$$\delta^t = \Gamma\Big[R^t + \text{sg}\big(V_{D_\phi}(S^{t+1})\big)\Big] - V_{D_\phi}(S^t),$$

can be efficiently computed via sparse matrix–vector multiplication. This yields the update

$$\phi \leftarrow \phi - c_v \nabla_\phi \big(n^{-1}\|\delta^t\|^2\big), \qquad (E.2)$$

where $c_v > 0$ denotes the critic learning rate.

**Actor update.** Given diffusion advantages $\hat{G}_{D_\phi}^t$, we update the policy parameters $\theta$ by maximising a surrogate objective of the form

$$J(\theta) = M^{-1} \sum_{t=0}^{M-1} J_t(\theta),$$

where the per-time-step surrogate objective is

$$\begin{aligned} J_t(\theta) = \ & c_r \sum_{i \in \mathcal{V}} \log \pi_{\theta_i}(A_i^t \mid \tau_i^t)\,(\hat{G}_{D_\phi}^t)_i \\ & + c_h \sum_{i \in \mathcal{V}} H\big[\pi_{\theta_i}(\cdot \mid \tau_i^t)\big]. \end{aligned} \qquad (E.3)$$

Here $H\big[\pi_{\theta_i}(\cdot \mid \tau_i^t)\big]$ denotes the entropy of agent $i$'s policy and $c_h \geqslant 0$ is an entropy coefficient. The first term in (E.3) corresponds to a weighted policy-gradient objective with scaling $c_r > 0$, where the diffusion advantage $(\hat{G}_{D_\phi}^t)_i$ acts as the training signal for agent $i$. As with standard actor–critic methods, we treat $\hat{G}_{D_\phi}^t$ as constant with respect to $\theta$ when computing $\nabla_\theta J_t$. The policy update is then given by

$$\theta \leftarrow \theta + c_j \nabla_\theta J(\theta), \qquad (E.4)$$

where $c_j > 0$ is the actor learning rate.

**Single-loop training.** In contrast to standard actor–critic implementations which first collect roll-outs and then perform optimisation, DA2C computes the diffusion advantages and TD residuals online during roll-out. This avoids

storing roll-outs and is well suited to our settings where the state distribution and observation statistics may drift over time (e.g., due to random walks in environment parameters). A limitation of this online approach is that we do not use an RNN-based critic, since we do not backpropagate through time over stored trajectories. Algorithm 1 summarises the procedure.

**Implementation details.** We use separate learning rates $c_v$ and $c_j$ for critic and actor updates in (E.2) and (E.4), respectively. We stop gradients through the bootstrap term $V_{D_\phi}(S^{t+1})$ when updating $\phi$. Further application-dependent modifications to the surrogate gain (e.g., regularisers and constraints) are introduced in Section E.5 for the LD-GNN network.

### E.5. LD-GNN network architectures

We describe the neural network architectures used in our application tasks. Unless stated otherwise, all multilayer perceptrons (MLPs) have two hidden layers with ReLU activations. For the LD-GNN actor, the message-aggregation module is a graph attention network (GAT) with dynamic attention (Brody et al., 2022), which we use for all tasks.

For the critic, we use feed-forward networks that take node and edge memory states as inputs (without parameter sharing with the actor). Although this is less expressive than a recurrent critic (Hausknecht and Stone, 2015), it enables efficient policy evaluation updates and improves training speed in practice. The number of message-passing layers determines the spatial propagation radius in critics. For the DVF and other local critics, we found that relatively shallow architectures already capture the most relevant dependencies due to the built-in spatial attenuation, while deeper models provide diminishing returns and may increase variance. This aligns with the intended behaviour of DVF, where influence decays with distance. In contrast, MAA2C benefits more from a larger critic radius, as it aims to approximate a global value function. However, even with an expanded critic, we observed that MAA2C does not match the performance of DA2C. This is because, even with an accurate estimate of the global value, it does not explicitly decompose agent contributions, leading to higher-variance policy gradient estimates at the individual agent level.

**Vector graph colouring.** We use node-memory dimension $\dim(X) = 32$ and omit edge memories $E$. For the DA2C critic, we use a five-layer graph isomorphism network (GIN) (Xu et al., 2019) which takes node and edge features as input, and proceed as in Section 5.2.1.

**Transmit power control.** We use $\dim(X) = \dim(E) = 10$. For the DA2C critic, we estimate the diffusion value

for each edge agent $(i, j)$ using

$$(V_{D_\phi})_{ij} = \mathrm{MLP}^V_\phi(X_i, E_{ij}, X_j),$$

as described in Appendix C.3.5. We also experimented with critics using wider neighbourhood context, but did not observe performance improvements.

We train node outputs $Y$ in an unsupervised manner by including the environment reward $r$ in the surrogate gain, thereby providing a direct learning signal for $\psi_\theta$. The message-passing policy $\lambda_\theta$ is trained using a diffusion-gradient term. To avoid the degenerate local optimum $\lambda_\theta \equiv 0$, we add a message-passing bonus $c_m|C|$ during early training, where $|C| = \sum_{i \in \mathcal{V}} |C_i|$ counts the total number of messages passed. The coefficient $c_m$ is annealed towards 0 over training iterations. This yields the surrogate gain

$$J(\theta) = \mathbb{E}_{\pi_\theta} \left[ r + c_r \sum_{(i,j) \in \mathcal{E}} \log \lambda_\theta(C_{ij}^t \mid I_i^t, E_{ij}^t)\, (\hat{G}_{D_\phi}^t)_{ij} \right.$$
$$\left. + c_h \sum_{(i,j) \in \mathcal{E}} H\big[\lambda_\theta(\cdot \mid I_i^t, E_{ij}^t)\big] + c_m|C| \right],$$

estimated as an empirical average over $M$ time steps.

**Optimisation details.** We apply the DA2C algorithm 1 to the task-dependent surrogate gain. We use batch size 64. For transmit power we use rollout length $M = 5$ and $N = 500$ training iterations; for graph colouring we use $M = 10$ and $N = 3000$. All models are trained using the Adam optimiser on a single GTX 1080 GPU. We tune the coefficients in the surrogate gain via grid search, and report the optimal values in Table 3, where $c_j$ and $c_v$ denote the learning rates (E.4), (E.2) for $\theta$ and $\phi$, respectively, and $c_h, c_m$ and $c_r$ are scalings in the surrogate gain. Experiments are implemented in PyTorch, and GNN components use PyG (Fey and Lenssen, 2019).

*Table 3.* Optimal learning rates and scalings for the LD-GNN network in the DA2C algorithm.

| Task | $c_j$ | $c_v$ | $c_r$ | $c_h$ | $c_m$ |
|---|---|---|---|---|---|
| Transmit power | 0.004 | 0.002 | 0.1 | 1e-4 | 0.2 |
| Graph colouring | 1 | 6e-4 | 5e-4 | 0 | 0 |

## F. Experimental details and additional results

### F.1. Comparison to classical baselines

Classical distributed graph colouring (DGC) algorithms typically assume free communication and do not explicitly account for a message-passing penalty $p_m$, which is central

---

**Algorithm 1** Single-loop DA2C training

---

Initialise actor parameters $\theta$ and critic parameters $\phi$
**for** policy iteration $= 1, \ldots, N$ **do**
    $J \leftarrow 0$
    Reset environment, obtain initial state $S^0$ and set $\tau^0$
    **for** $t = 0, \ldots, M - 1$ **do**
        Execute actions $A^t \sim \pi_\theta(\cdot \mid \tau^t)$, observe $(S^{t+1}, R^t)$ and update $\tau^{t+1}$
        Compute diffusion advantage $\hat{G}^t_{D_\phi}$ using (4.1)
        $J_t \leftarrow c_r \sum_{i \in \mathcal{V}} \log \pi_{\theta_i}(A^t_i \mid \tau^t_i)\,(\hat{G}^t_{D_\phi})_i + c_h \sum_{i \in \mathcal{V}} H[\pi_{\theta_i}(\cdot \mid \tau^t_i)]$
        $J \leftarrow J + J_t$
        $\delta^t \leftarrow \Gamma\big[R^t + \mathrm{sg}\big(V_{D_\phi}(S^{t+1})\big)\big] - V_{D_\phi}(S^t)$
        $\phi \leftarrow \phi - c_v \nabla_\phi \big(n^{-1}\|\delta^t\|^2\big)$                 (semi-gradient; stop-grad on $V_{D_\phi}(S^{t+1})$)
    **end for**
    $J \leftarrow M^{-1} J$
    $\theta \leftarrow \theta + c_j \nabla_\theta J$
**end for**

---

to our setting. To provide a representative hand-designed baseline, we consider a simple greedy local-update heuristic that trades off colouring quality with communication cost.

At each iteration $t$, we sample an active set of nodes $S^t \subset \{1, \ldots, n\}$ by including each node independently with probability $1/2$. Each active node $i \in S^t$ then updates its decision $Y^t_i$ using only information available from its neighbourhood. In our binary colouring formulation, this reduces to the threshold rule

$$Y^t_i = \begin{cases} 1, & \text{if } 2\,p_m \sum_{j \in N_i} Y^{t-1}_j < 1, \\ 0, & \text{otherwise,} \end{cases}$$

which allocates each agent's colours to maximise the sum of itself and its neighbour's rewards.

This greedy baseline yields reasonable solutions given sufficient iterations for all $p_m$ values considered, but is consistently outperformed by DA2C across the full penalty sweep (Figure 2).

We do not include an analogous greedy baseline for the transmit power optimisation tasks, as these involve continuous, strongly coupled control variables and non-separable reward terms, for which simple local threshold updates are ineffective.

### F.2. Barabási–Albert graph colouring results

We report additional graph colouring results on Barabási–Albert (BA) graphs, a standard scale-free graph model often used to capture degree heterogeneity and hub structure in real-world networks (Albert and Barabási, 2002). Compared to Erdős–Rényi (ER) graphs, BA graphs contain a small number of high-degree hub nodes, leading to shorter effective graph diameters and stronger coupling between agents.

Figure 7 shows training curves, and Figure 6 reports test performance across message-passing penalties $p_m$. DA2C continues to achieve the best performance on BA graphs, although the performance gap to baselines is smaller than on ER graphs. A plausible explanation is that hub nodes reduce the effective sparsity of the influence structure: because many agents are connected (directly or indirectly) through hubs, information propagates rapidly and neighbourhoods overlap heavily, making the task less locally decomposable.

**Effect of hub structure on diffusion value factorisation.** The diffusion value function exploits locality by weighting rewards according to both temporal distance and graph distance. In BA graphs, hub nodes cause the diffusion neighbourhood $N^t_i$ of many agents to expand quickly with diffusion time $t$, reducing the sparsity that DA2C leverages for structured value estimation. As a result, the advantage of a diffusion-based critic over global-critic baselines is attenuated, and methods that estimate global values from local information (e.g., MAA2C) exhibit performance closer to DA2C. In the limiting case of a fully connected graph, diffusion neighbourhoods cover the entire system immediately and DA2C reduces to MAA2C. Conversely, for graph families with slower neighbourhood growth—such as sparse ER graphs and many geometric graphs—DA2C benefits more strongly from sparsity, consistent with our empirical results.

### F.3. Out-of-distribution generalisation

To evaluate generalisation beyond the training distribution, we consider out-of-distribution (OOD) settings in which policies are tested on graph families and sizes not seen

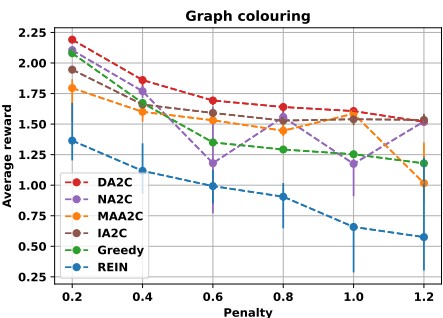

*Figure 6.* Test performance as a function of the message-passing penalty $p_m$ for Barabási-Albert graph colouring. Error bars indicate the lower and upper quartiles across runs.

achieves the highest returns after several hundred training iterations and exhibits stable convergence.

during training. Note that, in all experiments, training and evaluation already use independently sampled random graph instances, so test graphs are unseen even in the in-distribution setting.

We assess OOD generalisation on the graph colouring task. All models are trained on Erdős–Rényi (ER) graphs with $n = 5,000$ nodes, and tested on (i) larger ER graphs with $n = 10,000$ nodes and (ii) Barabási–Albert (BA) graphs with $n = 1,000$ nodes. Figure 8 reports test performance across message-passing penalties $p_m$.

Across both OOD settings, DA2C generalises robustly: it achieves the best performance across the full penalty sweep on the larger ER graphs, and remains top-performing on BA graphs for all but the largest penalty value. At the largest penalty, performance differences narrow and all methods approach the no-communication regime.

The performance on the larger 10,000-node ER graphs closely matches that on the training distribution, indicating robust scaling to larger graph sizes. Interestingly, performance on BA graphs is even higher than on ER graphs, and exceeds the performance obtained when training directly on BA graphs (cf. Figure 6). A possible explanation is that ER training provides a more homogeneous and stable learning distribution, acting as an implicit regulariser and reducing overfitting to high-degree hubs. In contrast, training on BA graphs can be dominated by hub nodes, which may increase gradient variance and encourage policies that specialise to hub-centric interaction patterns. The resulting ER-trained policies may therefore transfer better to BA graphs, where hubs reduce the effective graph diameter and facilitate rapid information propagation at test time.

### F.4. Firefighting training curves

For completeness, Figure 9 shows training curves for the firefighting task, illustrating convergence behaviour and variability across seeds for DA2C and the baselines. DA2C

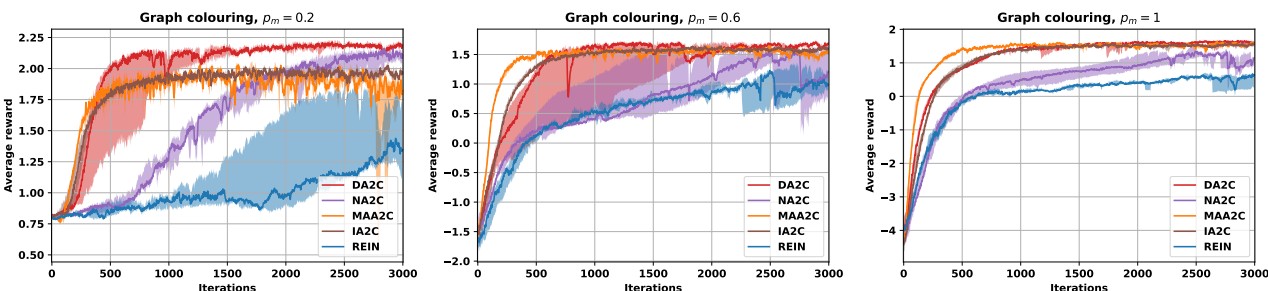

*Figure 7.* Training curves for different message-passing penalties $p_m$ on Barabási-Albert graph colouring. Error bars indicate the lower and upper quartiles across runs. Curves are smoothed with a moving average for visual clarity.

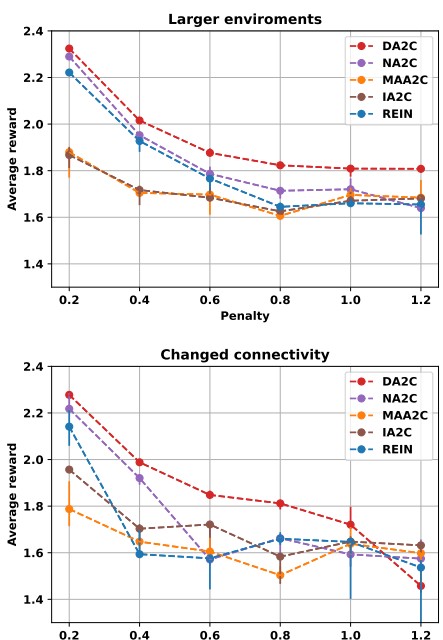

*Figure 8.* OOD generalisation on graph colouring as a function of message-passing penalty $p_m$. Models are trained on ER graphs ($n = 5{,}000$) and evaluated on larger ER graphs ($n = 10{,}000$) and BA graphs ($n = 1{,}000$). Error bars indicate lower and upper quartiles.

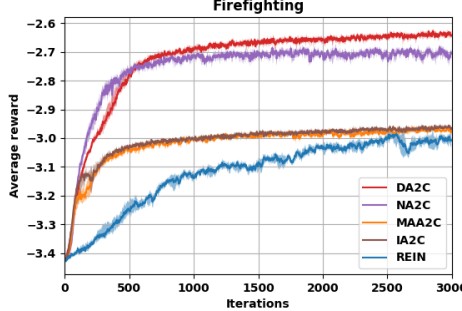

*Figure 9.* Training curves on the firefighting task. Shaded regions indicate the lower and upper quartiles across runs.

