# OpenReview forum: "Factored Value Functions for Graph-Based Multi-Agent Reinforcement Learning"
_ICML.cc/2026/Conference — ICML 2026 regular_

### Official Review · Reviewer_vGNn · 2026-03-06

**Soundness:** 3
**Presentation:** 2
**Significance:** 3
**Originality:** 2
**Overall Recommendation:** 4
**Confidence:** 3

**Summary:**

This paper proposes a new factored value function for graph-based multi-agent reinforcement learning (MARL), termed Diffusion Value Function (DVF). The method models credit propagation across an influence graph by combining temporal discounting with spatial attenuation. The authors show that DVF admits a Bellman fixed point and that the average of agent-wise DVF components recovers the global discounted value. Based on DVF, the paper further proposes Diffusion A2C (DA2C) and introduces a Learned DropEdge GNN (LD-GNN) actor to enable sparse communication. Experiments on the firefighting benchmark and several distributed computation tasks demonstrate performance improvements over local and global critic baselines.

**Compliance With Llm Reviewing Policy:**

Affirmed.

**Final Justification:**

The authors' response has clarified most of my questions. I have upgraded my evaluation accordingly.

**Key Questions For Authors:**

Compared with existing methods, how does the proposed method scale in terms of training and inference complexity? In particular, does the introduction of LD-GNN significantly increase the computational overhead?

**Limitations:**

yes

**Strengths And Weaknesses:**

**Strengths**:
1. The paper introduces the Diffusion Value Function (DVF), which addresses the middle ground between global value functions (suffer from weak credit assignment) and purely local ones (unstable in large-scale or infinite-horizon settings). The idea of explicitly modeling the "spatial decay of influence" via a diffusion operator is intuitive.
2. The proofs for the existence and uniqueness of the DVF as a fixed point of the modified Bellman operator are rigorous.
3. The experimental results consistently outperform several baseline methods (MAA2C, IA2C, and NA2C) across a variety of tasks, suggesting the potential effectiveness of the proposed approach.

**Weaknesses**:
1. The construction of the DVF relies heavily on a pre-defined and static influence graph, which may limit its applicability in environments where interaction structures are dynamic or unknown.
2. Although the paper compares DA2C with standard A2C and several of its variants, many state-of-the-art factored MARL methods are not included in the comparison, making it difficult to assess whether DVF provides a significant advantage over existing approaches.
3. The presentation in Section 5 and Section 6 could be improved to enhance the logical flow and clarity
4. The paper lacks a sufficiently comprehensive related work discussion. Incorporating and discussing more recent literature would help better position the contribution of this work relative to existing studies.

---

> ### Author Rebuttal · Authors · 2026-03-30
>
> We thank the reviewer for their thorough and constructive feedback and address each point below.
>
> **Influence graph assumption.**
>
> While the theoretical guarantees of DVF assume a known and static influence graph, the method described in the paper can be applied with approximate interaction structures, which extends its reach and application. This is demonstrated in the paper in the transmit power control experiments (Appendix C.3.5), where the influence graph is simplified, and we still see that DA2C still achieves strong empirical performance. More generally, DVF only requires a graph capturing interaction structure, which can be estimated in practice. Extending the method to fully dynamic or learned graphs is an interesting direction for future work.
>
> **MARL baselines.**
>
> Our goal was to isolate the contribution of the proposed critic (DVF), and therefore we focused on A2C-based baselines with a shared actor architecture. This enables a controlled comparison where performance differences can be attributed directly to the value function factorisation. The DVF is modular and can be combined with other MARL approaches.
>
> Many factored MARL methods (e.g., Sunehag et al., 2018; Rashid et al., 2020) assume fixed numbers of agents and feed-forward architectures, extending them to variable-agent graph-based settings with GNNs is typically non-trivial. Additionally, these methods focus on factorising critic networks, whereas our approach factorises the underlying MDP structure. This distinction is discussed in Appendix B.3, and we will make it more explicit in the main text.
>
> **Paper structure.**
>
> We agree that the presentation of Sections 5 and 6 can be improved. In particular, separating the application description from the methodological discussion can make the flow harder to follow. We will revise the structure so that the motivation and derivation of LD-GNN are presented more directly alongside the corresponding applications, improving both the clarity and the logical flow.
>
> **Computational complexity.**
>
> DA2C is designed to be scalable. Both the actor and critic rely on standard message-passing GNN operations, with computational cost that scales with the number of edges per layer. The diffusion operator $\Gamma$ can be implemented efficiently via sparse matrix--vector multiplication.
>
> LD-GNN does not significantly increase computational cost at inference time, and can in fact reduce it by sparsifying the communication graph before message passing. During training, it introduces additional memory overhead due to storing edge features for time-unrolling, which is typical for recurrent GNN architectures. Core computations such as the TD error (Eq. (3.5)) and advantage estimates (Eq. (4.1)) are fully vectorised and efficient on GPUs.
>
> **Related work discussion.**
>
> We agree that the related work section can be expanded. Whilst we cover the most relevant literature on Graph-MDPs, this area remains relatively under explored in reinforcement learning, with much of the existing work focusing on planning settings (e.g., Amato \& Oliehoek, 2015; Nair et al., 2005). In contrast, much of the research on general MARL methods (e.g., Wang \& Ke, 2022; Boehmer et al., 2020) focuses on unstructured environments and general-purpose coordination mechanisms. Our work instead targets settings with explicit interaction structure captured by a Graph-MDP, enabling value estimation that explicitly models spatial credit propagation. We will incorporate additional recent MARL work and clarify these relationships more explicitly in the revised manuscript.

---

> > ### Author Rebuttal · Reviewer_vGNn · 2026-04-03
> >
> > The authors' response has clarified most of my questions. I have upgraded my evaluation accordingly.

---

### Official Review · Reviewer_zRe5 · 2026-03-12

**Soundness:** 3
**Presentation:** 3
**Significance:** 3
**Originality:** 3
**Overall Recommendation:** 4
**Confidence:** 2

**Summary:**

1 	This paper studies cooperative multi-agent reinforcement learning on graph-structured systems, where agents interact primarily through local neighborhoods defined by an influence graph. The main contribution is the introduction of a Diffusion Value Function (DVF), a factored critic that models how rewards propagate across the graph with both temporal discounting and spatial diffusion. The authors argue that this design provides a more informative and scalable credit-assignment signal than either purely global critics or simpler local critics.

2 	Building on DVF, the paper proposes Diffusion Advantage Actor-Critic (DA2C), which replaces the standard scalar advantage with agent-wise diffusion advantages derived from the DVF Bellman backup. The paper also introduces LD-GNN, a graph neural network actor that learns sparse communication patterns by deciding which message-passing edges to activate during decentralized execution.

3 	The paper provides theoretical analysis showing that the proposed DVF is well-defined in the infinite-horizon setting, is the unique fixed point of a Bellman operator, and recovers the standard global discounted value when averaged across agents. It also establishes an alignment result relating improvements in diffusion values to improvements in the global objective. Empirically, the method is evaluated on several graph-based cooperative tasks, including firefighting, vector graph colouring, and transmit power control, where the proposed approach consistently outperforms several A2C-style baselines based on independent, neighborhood, and global critics.

**Compliance With Llm Reviewing Policy:**

Affirmed.

**Final Justification:**

Thanks for providing the responses. I like the idea and the overall quality of the paper. I would like to keep my score in that I am slightly not very familiar with the topic. But I think the authors have made a good work.

**Key Questions For Authors:**

1： The theoretical guarantees of the proposed Diffusion Value Function rely on the assumption that the environment admits an exact GMDP factorization with a known and fixed influence graph. However, in some of the evaluated tasks (e.g., transmit power control), the paper introduces an approximate edge-GMDP reformulation rather than strictly satisfying this assumption. Could the authors clarify to what extent the theoretical results are expected to hold in such approximate settings? In particular, do the authors have empirical or theoretical evidence suggesting that DVF remains stable when the factorization assumption is only approximately satisfied? A clearer discussion of this point would help assess the robustness and practical applicability of the approach.

2: The actor update in DA2C is motivated by the alignment properties of the diffusion value rather than being derived as an explicitly unbiased estimator of the global policy gradient. Could the authors clarify the theoretical interpretation of this update rule? For example, under what conditions (if any) does the diffusion advantage correspond to a valid policy gradient direction for the global objective? Understanding this relationship would help clarify whether DA2C should be viewed as a theoretically grounded policy-gradient method or primarily as a surrogate optimization approach.

3: The experimental evaluation mainly compares DA2C with A2C-style baselines that differ in the critic formulation (independent, neighborhood, or global critics). While this isolates the effect of the diffusion critic, it would be helpful to understand how the method compares with more recent graph-based MARL or communication-learning approaches. Could the authors comment on how DA2C relates to or differs from these methods, and whether additional empirical comparisons are possible? Such comparisons would help clarify the relative advantages of the proposed approach.

4: The method introduces a graph-based critic as well as edge-level memory in the LD-GNN actor. Could the authors provide more detail on the computational and memory overhead of the proposed approach compared to standard actor–critic implementations? In particular, how does the training cost scale with the number of agents and edges in large graphs? Additional discussion or empirical measurements of scalability would help assess the practicality of the method for large-scale systems.

**Limitations:**

Yes

**Strengths And Weaknesses:**

Soundness

The paper is technically sound overall. The proposed Diffusion Value Function (DVF) is supported by theoretical analysis, including existence, uniqueness as a Bellman fixed point, and a decomposition property relating diffusion values to the global value. The experimental methodology is generally reasonable, with multiple tasks, training curves, and ablation studies that help illustrate the behavior of the method. However, the theoretical guarantees rely on strong assumptions such as exact GMDP factorization and a fixed influence graph. In addition, the actor update is motivated by DVF alignment rather than a strictly unbiased global policy gradient estimator. Broader empirical comparisons beyond A2C-style critics would further strengthen the evidence.


Presentation

The paper is generally well structured and the main narrative is clear. The progression from GMDP formulation to the limitations of local/global critics, and then to the DVF and DA2C method, is easy to follow. Figures and tables help illustrate the experimental results, and the appendices provide useful implementation and theoretical details. Nevertheless, the paper could more clearly separate the conceptual contribution (DVF/DA2C) from the architectural component (LD-GNN). A more explicit discussion of computational complexity and scalability would also improve clarity.

**Significance**

The paper addresses an important problem in cooperative multi-agent reinforcement learning: credit assignment in graph-structured systems. This setting is increasingly relevant for large-scale distributed systems where interactions propagate through networks. The proposed diffusion-based value formulation provides an interesting intermediate perspective between purely local and purely global critics. This idea could inspire further research in structured MARL and distributed decision-making. However, the practical impact may be somewhat limited by the assumptions of a known and relatively stable interaction graph. Demonstrating the method in more complex or realistic environments would further strengthen its significance.


**Originality**

The main originality lies in the formulation of the Diffusion Value Function, which explicitly incorporates graph diffusion into the value definition rather than only modifying the critic architecture. The paper also emphasizes the conceptual distinction between factored value functions and previous approaches that factorize critic networks. In addition, the integration of the diffusion critic with a learned communication actor (LD-GNN) provides a coherent system-level approach. Overall, the work introduces a reasonably novel perspective on value estimation in graph-based MARL.

---

> ### Author Rebuttal · Authors · 2026-03-30
>
> We thank the reviewer for their thoughtful and detailed feedback and address their points below.
>
> **Approximate GMDP factorisation.**
>
> The DVF still yields a valid value decomposition, in the sense of Proposition 3.3, even under approximate influence graphs, as its proof does not rely on the exact form of the factorised reward or transition functions. As a result, the DVF components remain well-defined and bounded, which we also observe empirically in the transmit power tasks. More generally, DVF can be viewed as operating on a surrogate graph and reward decomposition; in this case, the guarantees hold with respect to the approximate model. When the influence graph is significantly misspecified, performance may degrade, as the GNN critic cannot accurately capture reward dependencies, but we observe that the method remains stable under moderate approximation.
>
> **Interpretation of the actor update.**
>
> The reviewer is correct that the DA2C update should be interpreted as a surrogate objective rather than a true estimator of the global policy gradient. The DVF introduces a spatial bias towards short-range rewards that is not present in the global value in (2.2), so the resulting objectives are different when applied to individual agents. This is analogous to the use of discounted value functions as surrogates for average-reward objectives, where a bias is introduced to improve tractability and learning stability. In practice, we find that this bias improves credit assignment and leads to improved empirical performance.
>
>
> **MARL baselines and relation to prior work.**
>
> Our experiments focus on A2C-style baselines to isolate the effect of the proposed critic (DVF) under a shared actor architecture. This enables a controlled comparison where improvements can be attributed directly to the value function factorisation.
>
> Conceptually, DVF differs from prior graph-based MARL methods in that it defines a factored value function via a diffusion process over the influence graph, rather than relying solely on architectural factorisation or learned communication. While many existing approaches use graphs for function approximation or communication learning (e.g., Foerster et al., 2016), or consider alternative MDP graph structures (e.g., Hu et al., 2023), they do not explicitly model spatial credit propagation in the value function itself or exploit the same GMDP factorisation. DVF is complementary to these approaches and can in principle be integrated with alternative actor architectures or training schemes.
>
>
> **Computational and memory complexity.**
>
> Both the graph critic and LD-GNN actor are built from standard message-passing GNN operations, with computational cost that scales with the number of edges per layer. The diffusion operator $\Gamma$ can be implemented via sparse matrix--vector multiplication, making each application efficient. The main overhead arises from storing edge features when training LD-GNNs for time-unrolling, which increases memory usage in dense graphs. In our experiments, we were able to train all environments on a single consumer GPU (GTX 1080), suggesting that the method remains practical at the scales considered.
>
> **Presentation and clarity.**
>
> We thank the reviewer for the positive feedback on the overall structure and clarity of the paper. We agree that the distinction between the conceptual contribution (DVF/DA2C) and the architectural component (LD-GNN) can be made clearer, and that the discussion of computational complexity can be expanded. We will revise the presentation to emphasise this separation more explicitly and provide a clearer account of scalability. In particular, DVF defines the value function independently of the actor, while LD-GNN is one possible instantiation for learning sparse communication policies.
>
> **Influence graph assumption and scope.**
>
> We agree that DVF assumes a known, time-static influence graph and GMDP factorisation, which restricts applicability to structured domains. This is a deliberate design choice: many real-world systems (e.g., power grids, communication networks, traffic systems) naturally exhibit such structured interactions, motivating our focus. Importantly, DVF does not require a perfect graph and can be applied with approximate or estimated interaction structures, as demonstrated in the transmit power control experiments (Appendix C.3.5). Extending DVF to fully dynamic or learned graphs is an interesting direction for future work.

---

> > ### Author Rebuttal · Reviewer_zRe5 · 2026-04-04
> >
> > thanks for providing the responses. I like the idea and the overall quality of the paper. I would like to keep my score in that I am slightly not very familiar with the topic. But I think the authors have made a good work.

---

### Official Review · Reviewer_ZgV1 · 2026-03-12

**Soundness:** 3
**Presentation:** 3
**Significance:** 3
**Originality:** 2
**Overall Recommendation:** 4
**Confidence:** 3

**Summary:**

This paper focuses on multi-agent reinforcement learning (RL) settings with a given graph structure (i.e., a graph Markov decision process), and presents a graph neural network- (GNN-)based critic architecture that uses the graph structure for decomposing the global value function into per-agent value functions that not only decays over time (as is the case with regular temporal discounting), but also attenuates spatially, under the assumption that each agent’s impact on other agents’ value functions should decays over the connectivity graph hops. The authors show that the proposed “diffusion value function” (DVF) is consistent with the standard global value objective and aligned with the agents' policy outcomes. The critic is trained using temporal-difference (TD) error minimization, while at the same time, a decentralized agent-level actor is trained using multi-step diffusion advantage estimates from the critic, resulting in the diffusion advantage actor-critic (DA2C) algorithm. Alternatively, for distributed algorithms that support message passing among agents, a Learned DropEdge GNN policy is proposed, comprising a message-passing policy that subsamples neighbors for each agent and an output policy that produces agent outputs given the subgraph induced by the subsampled agents. Across various cooperative graph-based multi-agent RL benchmarks, DA2C is shown to outperform several baselines, highlighting the significance of the proposed DVF.

**Compliance With Llm Reviewing Policy:**

Affirmed.

**Final Justification:**

Given the paper's contribution to the field of multi-agent reinforcement learning and the authors' response to all reviewers' comments, I recommend acceptance.

**Key Questions For Authors:**

1. In Eq. (2.1), does the agent’s incoming neighborhood include the agent itself? I believe it should, but it is not clear based on the definition.
2. Could you provide an intuitive explanation on why Proposition 3.3 is true? $V_D$ assumes spatial attenuation, but that is not necessarily the case for the global value function $V$.
3. Could the approach be extended to multiple underlying graphs, where each graph encodes a different type of interaction among the agents (thereby inducing different spatial attenuation patterns via DVF)?
4. If I am not mistaken, the environments considered in the experiments are all dense-reward experiments. Do you have any results/hypothesis on how DA2C will perform in sparse-reward settings, which are generally much more challenging?
5. It seems that you have used a graph isomorphism network (GIN) for the critic in the graph coloring problem. Does the critic architecture/scale play any role in the transfer behavior of the policy (i.e., the results in Figure 8)?
6. Minor comment: The $H_{ij} = H_{ji}$ assumption in Section C.3.1 is not necessarily correct; the incoming and outgoing interference for a given Tx-Rx pair could be different in principle (see, e.g., [A,B]).

[A] Wu, Xinzhou, Saurabha Tavildar, Sanjay Shakkottai, Tom Richardson, Junyi Li, Rajiv Laroia, and Aleksandar Jovicic. "FlashLinQ: A synchronous distributed scheduler for peer-to-peer ad hoc networks." IEEE/ACM Transactions on networking 21, no. 4 (2013): 1215-1228.

[B] Yi, Xinping, and Giuseppe Caire. "Optimality of treating interference as noise: A combinatorial perspective." IEEE Transactions on Information Theory 62, no. 8 (2016): 4654-4673.

**Limitations:**

Yes.

**Strengths And Weaknesses:**

**Strengths**

- The paper is, for the most part, very well-written and easy to follow.
- The considered problem and the paper’s contributions are interesting and significant. The experiments showcase the benefits of the proposed method in a large variety of challenging environments with different structures.
- The scalability and transferability of the method to larger graphs are important and useful results.
- The authors provide in-depth description of the environments and details on the experiments that should be very helpful with reproducibility and follow-up research on this topic.

​**Weaknesses**

- There are many multi-agent RL settings where there is not a prior given/known structure that governs the interactions among the agents. The authors do mention this limitation and the fact that DVF falls back to the standard global value function, but I still wonder if there is a way to extend this method to potentially infer an influence structure and train the critic on that learned graph structure.
- I was slightly confused when reading through Sections 5 and 6 of the manuscript. The flow, in my opinion, is a bit broken here, where Section 5 discusses applications, Section 6 goes back to discuss methodology, and Section 7 presents results. I would highly suggest rearranging these sections, so that the application descriptions are immediately followed by the results. One option could be merging Sections 4 and 6, but other options are also possible.

---

> ### Author Rebuttal · Authors · 2026-03-30
>
> We thank the reviewer for their thoughtful feedback and address their points below.
>
> **Influence graph learning.**
>
> We agree that extending DVF to settings where the interaction structure is not known a priori is an interesting direction. One possible approach would be to jointly learn an influence graph (e.g., via attention or link prediction mechanisms) alongside the critic. In this case, DVF could be applied to the learned graph, although establishing theoretical guarantees would be more challenging. In the absence of a meaningful interaction structure, DVF reduces to a global value function, as discussed in Appendix E.2.5. We will clarify this discussion in the revised manuscript.
>
> **Paper structure.**
>
> We agree that the flow between Sections 5–7 can be improved. In particular, separating application descriptions from methodology may disrupt readability. We will revise the structure so that application descriptions are more directly followed by their corresponding results.
>
> **Neighbourhood definition.**
>
> Both the incoming and outgoing neighbourhoods of an agent include the agent itself. This follows from the assumption that the influence graph is self-connected (line 79).
>
> **Intuition for Proposition 3.3.**
>
> Proposition 3.3 follows from the fact that the diffusion operator redistributes rewards across the graph while preserving their total contribution under discounting. In particular, since $\Gamma$ is column-stochastic up to the discount factor $\gamma$, we have $\mathbf{1}^\top \Gamma = \gamma \mathbf{1}^\top$, ensuring that the total contribution is preserved over time. Intuitively, DVF spreads each local reward over its neighbourhood while maintaining the total value, which explains why averaging agent-wise DVFs recovers the global value.
>
>
> **Multiple interaction graphs.**
>
> The method can be extended to settings with multiple interaction graphs when different types of interactions are present. A natural formulation is to decompose the reward into multiple components, each associated with its own influence graph, and define a separate diffusion value for each component. These can then be combined to form the overall value estimate.
>
> **Sparse-reward settings.**
>
> We do not currently include experiments in sparse-reward environments. We expect DVF to be most effective when rewards exhibit spatial locality, as it introduces an inductive bias favouring short-range credit assignment. In settings with highly sparse rewards and long-range dependencies, this bias may be less beneficial. Investigating this behaviour is an interesting direction for future work.
>
> **Critic architecture and transfer.**
>
> We did not observe a strong dependence of transfer performance on the critic architecture. However, we found that GIN-based critics consistently outperform attention-based GNNs (GAT and GATv2) in both in-distribution and transfer settings.
>
> **Interference model.**
>
> We thank the reviewer for pointing this out. The matrix $H$ need not be symmetric (e.g., in frequency division duplex (FDD) systems), and we will clarify this in Appendix C.3.1. We note that $H_{ij}$ represents channel conditions rather than interference directly.

---

> > ### Author Rebuttal · Reviewer_ZgV1 · 2026-03-31
> >
> > Thank you for your rebuttal. I maintain my favorable rating for the paper.

---

### Official Review · Reviewer_k8F9 · 2026-03-13

**Soundness:** 3
**Presentation:** 3
**Significance:** 2
**Originality:** 3
**Overall Recommendation:** 4
**Confidence:** 3

**Summary:**

This paper addresses the credit assignment problem in large-scale cooperative MARL on Graph-based Markov Decision Processes (GMDPs), where state transitions and rewards factor according to a known influence graph. The central observation is that standard value functions occupy an awkward middle ground for this setting: global critics aggregate reward information from all agents and become increasingly insensitive to individual agents as systems grow; purely local critics, by contrast, suffer from unboundedness in infinite-horizon settings when a graph's average out-degree exceeds one.
To address this, the paper introduces the Diffusion Value Function (DVF), which assigns each agent a value component by propagating reward credit through the influence graph using a spatially and temporally attenuated diffusion operator Γ = γAD⁻¹. The DVF is shown to be well-defined (bounded, convergent), to admit a unique Bellman fixed point, to decompose the global discounted value via an averaging property, and to be policy-aligned in the sense that improving all diffusion components guarantees improvement in the global objective. These properties are proved, making DVF a theoretically grounded alternative to both local and global critics.
The authors instantiate DVF as the critic in Diffusion A2C (DA2C) and pair it with a novel actor architecture, the Learned DropEdge GNN (LD-GNN), which learns sparse communication decisions under message-passing costs. Experiments span firefighting, vector graph colouring, and two wireless transmit power control tasks across 30+ environment configurations. DA2C consistently outperforms local, global, and neighbourhood baselines, with strong out-of-distribution generalization to larger and structurally distinct graphs.

**Compliance With Llm Reviewing Policy:**

Affirmed.

**Key Questions For Authors:**

1.	Guarantee preservation under approximate GMDP: The decomposition and policy-alignment properties (Propositions 3.3 and 3.4) are proved for exact GMDP factorizations. In the transmit power control tasks, an approximate GMDP is used with simplified influence graphs and edge rewards derived from smoothed node rewards (Equation C.1). Do the authors claim that these guarantees hold approximately under this formulation, and if so, can they provide either a formal bound or an informal argument quantifying the approximation error? This seems important for understanding when DVF-based critics are safe to apply in practice.
2.	Spatial attenuation radius sensitivity: The number of GNN message-passing layers in the DVF critic determines how far reward information propagates when estimating diffusion values. Since the theoretical motivation for DVF explicitly involves spatial attenuation, it is natural to ask: how sensitive are results to this architectural choice? Does using a deeper critic (wider receptive field) close the gap with MAA2C, and does a very shallow critic (single layer) degrade performance substantially? An ablation here would directly test whether the diffusion structure is doing the expected work.
3.	Interaction between sparse actor and critic structure: When LD-GNN learns to drop edges, the effective communication graph during training can differ substantially from the static influence graph used by the DVF critic. Since the GMDP reward factorization and the diffusion operator Γ are defined relative to the influence graph, does this create a mismatch between what the critic is estimating and the actual dependency structure experienced by the actor? Have the authors observed any training instabilities or systematic biases that might be attributable to this mismatch?
4.	Comparison to Jing et al. (2024): The paper cites Jing et al. (2024, IEEE TAC) as a directly related method that also defines graph-induced local value functions for distributed MARL. This method is cited but not included as a baseline. Could the authors explain why this comparison was excluded, and whether it is feasible to include it? Given the theoretical and application overlap, a direct comparison would substantially strengthen the empirical claims.

**Limitations:**

yes

**Strengths And Weaknesses:**

****Strengths****

***Soundness*** The theoretical contribution is a clear strength of the paper. Propositions 3.1–3.4 cover the full chain of properties one would want from a surrogate value function: existence, uniqueness, decomposition, and policy alignment. The proofs, to my knowledge, are correct and self-contained. The key insight that the column-stochastic property of AD⁻¹ (Lemma B.1) ensures the diffusion operator is a contraction under the ℓ₁ operator norm is concise and elegant. The appendix example (Appendix A.2) that shows local values can diverge while DVF remains bounded for all γ ∈ (0,1) is a good illustration that makes the motivation for DVF concrete. The separation of DVF from factorized critic architectures such as QMIX and VDN (Appendix B.3) is also appreciated and overdue in the literature.

The experimental design is careful. All A2C variants (IA2C, NA2C, MAA2C, DA2C) share an identical actor and differ only in the critic, cleanly isolating the effect of the value function structure, a methodological discipline that is rarer than it should be in MARL papers. The ablation in Table 2 is informative. The OOD generalization study in Appendix F.3 (training on 5,000-node ER graphs and evaluating on 10,000-node ER and BA graphs) is a useful test, and the fact that ER-trained policies transfer surprisingly well to BA graphs prompts an interesting explanation about hub-induced regularization that the authors handle honestly rather than overselling.

***Originality*** While GMDP-based value factorization has appeared in planning literature (Oliehoek et al., 2008; Schneider et al., 1999), bringing it into the context of deep RL with GNN critics and provably valid Bellman operators is a meaningful step. The specific choice to use the row-normalized adjacency matrix as a diffusion operator, rather than the raw adjacency, is not obvious and is well-motivated. The connection to random-walk diffusion (Appendix B.4) positions DVF within a broader mathematical framework. The LD-GNN actor, which treats communication decisions as explicit learnable variables rather than attention weights, is a practical contribution distinct from prior message-passing attention approaches.

***Presentation*** The paper is well-written and logically structured. The notation is consistent and introduced carefully. The decision to prove limitations of the local value function (Appendix A.2) before introducing DVF creates a clean narrative arc. Section 3.5 (Limitations) is placed early and is substantive. The appendix is rather dense but navigable, with each section serving a clear purpose. Figures 2 and 3 together provide a good picture of both final performance and training dynamics.

****Weaknesses****

***Significance*** The method requires a known, time-static influence graph and a valid GMDP factorization with local rewards. This is a genuine restriction: most widely used MARL benchmarks (StarCraft II, MPE, SMAC) do not satisfy these structural assumptions, as the authors acknowledge in Appendix E.2.5. The paper justifies this by framing DVF as being most appropriate for structured infrastructure domains (power grids, communication networks, traffic). That is a fair framing, but it does limit the addressable audience. The paper would benefit from a more explicit discussion of whether approximate GMDP formulations, as used in the transmit power control tasks, preserve the theoretical guarantees of Propositions 3.3 and 3.4, or whether those properties become approximations.

The baselines are all variants of A2C distinguished solely by critic structure. This is an appropriate design for isolating the critic's effect, but it means the paper does not compare against other approaches designed specifically for large-scale GMDP-structured tasks, such as distributed value function methods with neighbor communication (e.g., Jing et al., 2024, which is cited but not compared against), or model-based approaches that exploit the known transition structure of GMDPs. A comparison to at least one such method would strengthen the claim that DVF-based critics are competitive beyond the set of generic A2C variants.

***Soundness*** The transmit power control tasks are handled via an approximate GMDP formulation (Appendix C.3.5), in which the influence graph is deliberately simplified to avoid excessive density. This is practically motivated, but the decomposition and policy-alignment guarantees (Propositions 3.3 and 3.4) are proved under the exact GMDP factorization. It is not discussed what happens to these guarantees under the approximation, or how the edge reward assignment via smoothed node rewards (Equation C.1) affects the relationship between the edge GMDP objective and the original node-level objective. This gap between theory and experiment is the most significant technical concern in the paper.

No hyperparameter sensitivity analysis is reported for the DVF-specific parameter, namely the number of GNN message-passing layers in the critic (which determines the receptive field and thus the effective spatial attenuation radius). Similarly, the message-passing bonus coefficient cm used in LD-GNN training is annealed from 0.2 to 0 over training; the sensitivity of results to this schedule is not discussed.


***Presentation*** The LD-GNN actor is presented as a contribution of comparable weight to DVF, but the integration between the two is somewhat loosely described. In particular, it is not made explicit whether the theoretical properties of DA2C (specifically, Propositions 3.3 and 3.4) continue to hold when the actor learns sparse communication: since sparse communication changes the effective GMDP structure during training, potentially causing the influence graph to become non-stationary. This connection deserves at least a paragraph in the main paper rather than being handled implicitly.

---

> ### Author Rebuttal · Authors · 2026-03-30
>
> We thank the reviewer for their constructive feedback and address each point below.
>
> **Influence graph assumption and scope.**
>
> We agree that DVF assumes a known, time-static influence graph and GMDP factorisation, which restricts applicability to structured domains. This is a deliberate design choice, as many real-world systems (e.g., power grids, communication networks, traffic systems) naturally exhibit such structured interactions, where modelling influence explicitly is beneficial for credit assignment. Importantly, DVF does not require a perfectly specified graph and can be applied with approximate or estimated interaction structures. This is demonstrated in the transmit power control experiments (Appendix C.3.5), where the graph is simplified yet DA2C still performs well. Extending DVF to fully dynamic or learned graphs is an interesting direction for future work.
>
> **Guarantees under approximate GMDP.**
>
> We agree this is an important point and appreciate the reviewer highlighting the gap between theory and approximate settings. Propositions 3.3 and 3.4 are derived under exact GMDP factorisation. In the approximate setting used in Appendix C.3.5, DVF can be interpreted as operating on a surrogate graph and reward decomposition (Eq. (C.1)). The guarantees then hold with respect to this approximate model. The deviation from the true DVF objective depends on the approximation error in both the influence graph and reward decomposition. While deriving formal bounds is an interesting direction for future work, empirically we observe that DVF remains robust under such approximations, suggesting graceful degradation when GMDP assumptions are only approximately satisfied.
>
> **Spatial attenuation radius (GNN depth).**
>
> The number of message-passing layers determines the spatial propagation radius in critics. For the DVF and other local critics, we find that relatively shallow architectures already capture the most relevant dependencies due to the built-in spatial attenuation, while deeper models provide diminishing returns and may increase variance. This aligns with the intended behaviour of DVF, where influence decays with distance.
>
> In contrast, MAA2C benefits more from a larger critic radius, as it aims to approximate a global value function. However, even with an expanded critic, we observe that MAA2C does not match the performance of DA2C. This is because, even with an accurate estimate of the global value, it does not explicitly decompose agent contributions, leading to higher-variance policy gradient estimates at the individual agent level.
>
> **Interaction between LD-GNN actor and DVF critic.**
>
> The underlying GMDP influence graph is independent of whether two nodes choose to communicate at a given time step. If an edge $(i,j)$ is dropped by the LD-GNN, node $i$ still influences node $j$ through this action, as the decision not to communicate is itself a form of interaction. Therefore, the influence structure that defines the DVF remains unchanged.
>
> As a result, LD-GNN sparsifies communication at the actor level without modifying the underlying environment interaction graph used by the DVF critic. The theoretical properties of DVF (Propositions 3.3 and 3.4) therefore continue to hold with respect to this underlying influence graph.
>
>
> **Comparison to Jing et al. (2024).**
>
> Jing et al.  (2024) considers value functions closely related to the local value function defined in Eq. (3.1), along with truncated variants similar to NA2C. As discussed in Appendix A.2, directly estimating such local value functions becomes increasingly challenging in larger environments due to their exponential growth with neighbourhood size. In addition, these formulations do not admit simple TD errors such as Eq. (3.5), which limits the scalability of critic learning in practice.
>
> Our work instead introduces a diffusion-based factorisation that preserves a tractable TD learning objective while capturing spatial credit propagation. The diffusion operator $\Gamma = \gamma A D^{-1}$ corresponds to a message-passing step and can be implemented via sparse matrix–vector multiplication, resulting in computational cost that scales with the number of edges per application. This enables the method to be applied to larger graphs in practice.

---

> > ### Author Rebuttal · Reviewer_k8F9 · 2026-04-03
> >
> > Thank you, authors. Issues I raised have been resolved. I stand by my initial positive assessment.

---

### Decision · Program_Chairs · 2026-04-30

**Decision:**

Accept (regular)

**Comment:**

This paper addresses the credit assignment problem in large scale cooperative MARL on Graph based Markov Decision Processes (GMDPs), where state transitions and rewards factor according to a known influence graph. The authors identify a fundamental trade off: global critics become insensitive to individual agents as systems grow, while purely local critics can be unbounded in infinite horizon settings when the average out degree exceeds one.

To bridge this gap, they introduce the Diffusion Value Function (DVF), which assigns each agent a value component by propagating reward credit through the influence graph using a spatially and temporally attenuated diffusion operator. The DVF is shown to be well defined (bounded, convergent), to admit a unique Bellman fixed point, to decompose the global discounted value via an averaging property, and to be policy aligned (improving all diffusion components guarantees improvement of the global objective).

Based on DVF, the authors propose Diffusion A2C (DA2C) and complement it with a Learned DropEdge GNN (LD GNN) actor that learns sparse communication patterns. Experiments on firefighting, vector graph colouring, and wireless transmit power control tasks (30+ configurations) show consistent outperformance over local, neighbourhood, and global A2C baselines, as well as strong out of distribution generalization to larger and structurally different graphs.

All reviewers are positive. The paper is technically sound, theoretically and empirically thorough. The main limitations (known static graph, approximate GMDP in one environment) are clearly stated. Several points need to be addressed:

* Add a brief sensitivity analysis for the number of GNN layers (spatial attenuation radius) or at least state the default choice and note that deeper layers give diminishing returns.
* Explicitly clarify that the DA2C actor update is a surrogate objective, not an unbiased policy gradient estimator.
* Include a short paragraph on computational complexity (scaling with edges, sparse matrix operations, memory overhead of LD‑GNN).